# EXPECTED PROBABILISTIC HIERARCHIES

## ABSTRACT

Hierarchical clustering has usually been addressed by discrete optimization using heuristics or continuous optimization of relaxed scores for hierarchies. In this work, we propose to optimize *expected* scores under a probabilistic model over hierarchies. **(1)** We show *theoretically* that the global optimal values of the expected Dasgupta cost and Tree-Sampling divergence (TSD), two unsupervised metrics for hierarchical clustering, are equal to the optimal values of their discrete counterparts contrary to some relaxed scores. **(2)** We propose Expected Probabilistic Hierarchies (EPH), a probabilistic model to learn hierarchies in data by optimizing expected scores. EPH uses *differentiable hierarchy sampling* enabling end-to-end gradient descent based optimization, and an *unbiased subgraph sampling* approach to scale to large datasets. **(3)** We evaluate EPH on synthetic and real-world datasets including *vector* and *graph* datasets. EPH outperforms all other approaches on quantitative results and provides meaningful hierarchies in qualitative evaluations.

## 1 INTRODUCTION

A fundamental problem in unsupervised learning is clustering. Given a dataset, the task is to partition the instances into similar groups. While *flat* clustering algorithms such as $k$-means group data points into disjoint groups, a *hierarchical* clustering divides the data recursively into smaller clusters, which yields several advantages over a flat one. Instead of only providing cluster assignments of the data points, it captures the clustering at multiple granularities, allowing the user to choose the desired level of fine and coarseness depending on the task. The hierarchical structure can be easily visualized in a *dendrogram* (e.g., see Fig. 4), making it easy to interpret and analyze. Hierarchical clustering finds applications in many areas, from personalized recommendation (Zhang et al., 2014) and document clustering (Steinbach et al., 2000) to gene-expression (Eisen et al., 1998) and phylogenetics (Felsenstein, 2004). Furthermore, the presence of hierarchical structures can be observed in many real-world graphs in nature and society (Ravasz & Barabási, 2003).

A first family of methods for hierarchical clustering are discrete approaches. They aim at optimizing some hierarchical clustering quality scores on a discrete search space, i.e.:

$$\max_{\hat{\mathcal{T}}} \text{score}(\boldsymbol{X}, \hat{\mathcal{T}}) \text{ s.t.} \hat{\mathcal{T}} \in \text{discrete hierarchies,} \tag{1}$$

where $\boldsymbol{X}$ denotes a given (vector or graph) dataset. Examples of score optimization could be the minimization of the discrete Dasgupta score Dasgupta (2016), the minimization of the error sum of squares (Ward Jr, 1963), the maximization of the discrete TSD Charpentier & Bonald (2019), or the maximization of the modularity score (Blondel et al., 2008). Discrete approaches have two main limitations: the optimization search space of discrete hierarchies is large and constrained, which often makes the problem intractable without using heuristics, and the learning procedure is not differentiable and thus not amenable to gradient-based optimization, as done by most deep learning approaches. To mitigate these issues, a second more recent family of continuous methods proposes to optimize some (soft-)scores on a continuous search space of relaxed hierarchies:

$$\max_{\mathcal{T}} \text{soft-score}(\boldsymbol{X}, \mathcal{T}) \text{ s.t.} \mathcal{T} \in \text{relaxed hierarchies.} \tag{2}$$

Examples are the relaxation of Dasgupta (Chami et al., 2020; Chierchia & Perret, 2019; Zügner et al., 2022) or TSD scores (Zügner et al., 2022). A major drawback of continuous methods is that the optimal value of soft scores might not align with their discrete counterparts.

**Contributions.** In this work, we propose to optimize *expected* discrete scores, called Exp-Das and Exp-TSD, instead of the *relaxed* soft scores, called Soft-Das and Soft-TSD (Zügner et al., 2022). In particular, our contributions can be summarized as follows:

- **Theoretical contribution**: We analyze the *theoretical* properties of both the soft and expected scores. We show that the minimal value of Soft-Das can be different from that of the discrete Dasgupta cost while the optimal values of the expected scores are equal to their optimal discrete counterparts.

- **Model contribution**: We propose a new method called Expected Probabilistic Hierarchies (EPH) to optimize the Exp-Das and Exp-TSD. EPH provides an *unbiased* estimate of Exp-Das and Exp-TSD with biased gradients based on differentiable hierarchy sampling. By utilizing an *unbiased* subgraph sampling procedure, EPH *scales* to large (vector) datasets.

- **Experimental contribution**: In quantitative experiments, we show that EPH outperforms other baselines on 20/24 cases on 16 datasets, including both *graph* and *vector* datasets. In qualitative experiments, we show that EPH provides *meaningful* hierarchies.

## 2 RELATED WORK

**Discrete Methods.** We further differentiate between agglomerative (bottom-up) and divisive (top-down) discrete algorithms. Well-established agglomerative methods are the linkage algorithms that subsequently merge the two clusters with the lowest distance into a new cluster. There are several ways to define the similarity of two clusters. The average linkage (AL) method uses the average similarity, while single linkage (SL) and complete linkage (CL) use the minimum and maximum similarity between the groups, respectively (Hastie et al., 2009). Finally, the ward linkage (WL) algorithm (Ward Jr, 1963) operates on Euclidean distances and merges the two clusters with the lowest increase in the sum of squares. Another agglomerative approach is the Louvain algorithm (Blondel et al., 2008) which maximizes iteratively the modularity score. A more recent parallelized agglomerative graph-based approach, ParHAC, enables scaling to massive datasets. In terms of quality, however, ParHAC performs inferior to AL w.r.t. the Dasgupta cost (Dhulipala et al., 2022). Unlike agglomerative methods, divisive algorithms work in a top-down fashion. Initially, all leaves share the same cluster and are recursively divided into smaller ones using flat clustering algorithms. Famous examples are based on the $k$-means algorithm (Steinbach et al., 2000) or use approximations of the sparsest cut (Dasgupta, 2016).

**Continuous Methods.** In recent years, many continuous algorithms emerged to solve hierarchical clustering. These methods minimize continuous relaxations of the Dasgupta cost using gradient descent based optimizers. Monath et al. (2017) optimized a probabilistic cost version. To parametrize the probabilities, they performed a softmax operation on learnable routing functions from each node on a fixed binary hierarchy. Chierchia & Perret (2019) proposed UFit, a model operating in the ultra-metric space. Furthermore, to optimize their model, they presented a soft-cardinal measure to compute a differentiable relaxed version of the Dasgupta cost. Other approaches operate on continuous representations in hyperbolic space, such as gHHC (Monath et al., 2019) and HypHC (Chami et al., 2020). Zügner et al. (2022) recently presented a flexible probabilistic hierarchy model (FPH) on which our method is based. FPH directly parametrizes a probabilistic hierarchy and substitutes the discrete terms in the Dasgupta cost and Tree-Sampling Divergence with their probabilistic counterparts. This results in a differentiable objective function which they optimize using projected gradient descent.

**Differentiable Sampling Methods.** Stochastic models with discrete random variables are difficult to train as the backpropagation algorithm requires all operations to be differentiable. To address this problem, estimators such as the Gumbel-Softmax (Jang et al., 2016) or Gumbel-Sinkhorn (Mena et al., 2018) are used to retain gradients when sampling discrete variables. These differentiable sampling methods have been used for several tasks, including DAG predictions (Charpentier et al., 2022), spanning trees or subset selection (Paulus et al., 2020), and generating graphs Bojchevski et al. (2018). In contrast to prior work, EPH enables optimizing the expected clustering scores under a probabilistic hierarchy by utilizing differentiable sampling. Note that sampling spanning trees is not applicable in our case since we have a restricted structure where the nodes of the graph correspond to the leaves of the tree.

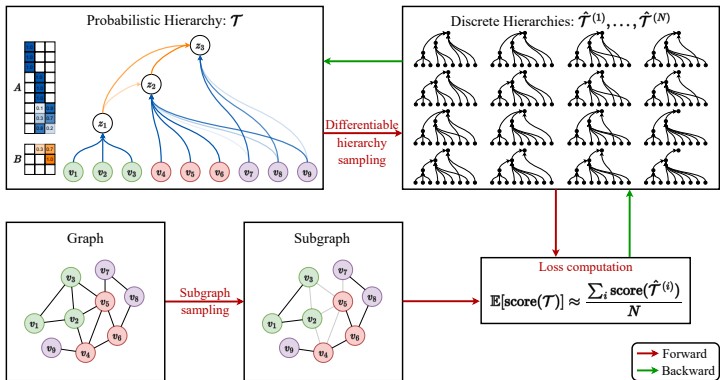

Figure 1: Overview of our proposed EPH model. A formal description is given in App. B.9.

## 3 PROBABILISTIC HIERARCHICAL CLUSTERING

We consider a graph dataset. Let $\mathcal{G} = (V, E)$ be a graph with $n$ vertices $V = \{v_1, \ldots, v_n\}$ and $m$ edges $E = \{e_1, \ldots, e_m\}$. Let $w_{i,j}$ denote the weight of the edge connecting the nodes $v_i$ and $v_j$ if $(i, j) \in E$, 0 otherwise, and $w_i = \sum_j w_{i,j}$ the weight of the node $v_i$. In general, we assume a directed graph, i.e. $w_{i,j} \neq w_{j,i}$. We define the edge distribution $P(v_i, v_j)$ for pairs of nodes, $P(v_i, v_j) \propto w_{i,j}$, s.t. $\sum_{v_i, v_j \in V} P(v_i, v_j) = 1$ and equivalently the node distribution $P(v_i) \propto w_i$, s.t. $\sum_{v_i \in V} P(v_i) = 1$. We can extend this representation to any vector dataset $\mathcal{D} = \{x_1, \ldots, x_n\}$ and interpret the dataset as a graph by using the data points $x_i$ as nodes and pairwise similarities (e.g., cosine similarities) as edge weights.

**Discrete hierarchical clustering.** We define a discrete hierarchical clustering $\hat{\mathcal{T}}$ of a graph $\mathcal{G}$ as a rooted tree with $n$ leaves and $n'$ internal nodes. The leaves $V = \{v_1, v_2, \ldots, v_n\}$ represent the nodes of $\mathcal{G}$, while the internal nodes $Z = \{z_1, z_2, \ldots, z_{n'}\}$ represent clusters, with $z_{n'}$ being the root node. Each internal node groups the data into disjoint sub-clusters, where edges reflect memberships of clusters. We can represent the hierarchy using two binary adjacency matrices $\hat{A} \in \{0, 1\}^{n \times n'}$ and $\hat{B} \in \{0, 1\}^{n' \times n'}$, i.e., $\hat{\mathcal{T}} = (\hat{A}, \hat{B})$. While $\hat{A}$ describes the edges from the leaves to the internal nodes, $\hat{B}$ specifies the edges between the internal nodes. Since every node in the hierarchy except the root has exactly one outgoing edge, we have the following constraints: $\sum_j^{n'} \hat{A}_{i,j} = 1$ for $1 \leq i \leq n$, $\sum_j^{n'} \hat{B}_{i,j} = 1$ for $1 \leq i < n'$ and $\sum_j^{n'} \hat{B}_{n',j} = 0$ for the last row. Thus, except for the last row of $\hat{B}$, both matrices are row-stochastic. We denote the ancestors of $v$ as $\mathrm{anc}(v)$, and the lowest common ancestor (LCA) of the two leaves $v_i$ and $v_j$ in $\hat{\mathcal{T}}$ as $v_i \wedge v_j$.

**Probabilistic hierarchical clustering.** Zügner et al. (2022) recently proposed probabilistic hierarchies. The idea is to use a continuous relaxation of the binary adjacency matrices while keeping the row-stochasticity constraints. Thus, we end up with two matrices $A \in [0, 1]^{n \times n'}$ and $B \in [0, 1]^{n' \times n'}$. The entries represent parent probabilities, i.e., $A_{i,j} := p(z_j | v_i)$ describes the probability of the internal node $z_j$ being the parent of $v_i$ and $B_{i,j} := p(z_j | z_i)$ the probability of the internal node $z_j$ being the parent of $z_i$. Together, they define a probabilistic hierarchy $\mathcal{T} = (A, B)$. Given such a probabilistic hierarchy, one can easily obtain a discrete hierarchy by interpreting the corresponding rows of $A$ and $B$ as categorical distributions. We sample an outgoing edge for each leaf and internal node. Since $B$ is restricted to be an upper triangular matrix, this tree-sampling procedure will result in a valid discrete hierarchy, denoted by $\hat{\mathcal{T}} = (\hat{A}, \hat{B}) \sim P_{A,B}(\mathcal{T})$.

## 4 EXPECTED PROBABILISTIC HIERARCHICAL CLUSTERING

### 4.1 EXPECTED METRICS

Unlike flat clusterings, there has been a shortage of objective functions for hierarchical clusterings. Thus, many algorithms to derive hierarchies were developed without a precise objective. An objective function not only allows us to evaluate the performance of a hierarchy but also yields possibilities for optimization techniques. Recently the two unsupervised functions Dasgupta cost (**Das**) (Dasgupta,

2016) and Tree-Sampling Divergence **(TSD)** (Charpentier & Bonald, 2019) were proposed, triggering the development of a new generation of hierarchical clustering algorithms. The Dasgupta cost is a well-established metric for graphs and vector data, while the TSD is a recent metric specifically tailored to graphs. In addition to being unsupervised, i.e., applicable in cases where the data is unlabeled, both metrics also have intuitive motivations. The metrics can be written as:

$$\text{Das}(\hat{\mathcal{T}}) = \sum_{\mathbf{v}_i, \mathbf{v}_j \in V} P(\mathbf{v}_i, \mathbf{v}_j) c(\mathbf{v}_i \wedge \mathbf{v}_j) \qquad \text{and} \qquad \text{TSD}(\hat{\mathcal{T}}) = \text{KL}(p(\mathbf{z}) \| q(\mathbf{z})), \tag{3}$$

where $c(\mathbf{z})$ is the number of leaves whose ancestor is z, i.e., $c(\mathbf{z}) = \sum_{\mathbf{v}_i \in V} \mathbf{1}_{[\mathbf{z} \in \text{anc}(\mathbf{v}_i)]}$, and $p(\mathbf{z})$ and $q(\mathbf{z})$ are two distributions induced by the edge and node distributions, i.e., $p(\mathbf{z}) = \sum_{\mathbf{v}_i, \mathbf{v}_j} \mathbf{1}_{[\mathbf{z} = \mathbf{v}_i \wedge \mathbf{v}_j]} P(\mathbf{v}_i, \mathbf{v}_j)$ and $q(\mathbf{z}) = \sum_{\mathbf{v}_i, \mathbf{v}_j} \mathbf{1}_{[\mathbf{z} = \mathbf{v}_i \wedge \mathbf{v}_j]} P(\mathbf{v}_i) P(\mathbf{v}_j)$. Dasgupta favors similar leaves to have the lowest common ancestor low in the hierarchy (Dasgupta, 2016). TSD quantifies the ability to reconstruct the graph from the hierarchy in terms of information loss Charpentier & Bonald (2019). Hence, any hierarchy achieving a good score provides a good compression of the original graph. In practice, both metrics are good indicators of meaningful hierarchies (Dasgupta, 2016; Charpentier & Bonald, 2019; Chami et al., 2020; Zügner et al., 2022). Recently, Zügner et al. (2022) proposed the Flexible Probabilistic Hierarchy (FPH) method. FPH substitutes the indicator functions with their corresponding probabilities under the tree-sampling procedure, obtaining cost functions for probabilistic hierarchies called **Soft-Das** and **Soft-TSD**. These two metrics correspond to the scores of the *expected hierarchies* under the tree sampling procedure (see App. A.1). In contrast, we propose to optimize the *expected metrics* in this work. Intuitively, this corresponds to moving the expectation from inside the metric functions to outside, reflecting the natural way of performing Monte-Carlo approximation via (tree-) sampling. More specifically, our objectives are:

$$\min_{\boldsymbol{A}, \boldsymbol{B}} \mathbb{E}_{\hat{\mathcal{T}}} \left[ \text{Das}(\hat{\mathcal{T}}) \right] \text{ s.t. } \hat{\mathcal{T}} \sim P_{\boldsymbol{A}, \boldsymbol{B}}(\mathcal{T}) \qquad \text{and} \qquad \max_{\boldsymbol{A}, \boldsymbol{B}} \mathbb{E}_{\hat{\mathcal{T}}} \left[ \text{TSD}(\hat{\mathcal{T}}) \right] \text{ s.t. } \hat{\mathcal{T}} \sim P_{\boldsymbol{A}, \boldsymbol{B}}(\mathcal{T}), \tag{4}$$

which we denote as **Exp-Das** and **Exp-TSD**. Note that we optimize over $\boldsymbol{A}$ and $\boldsymbol{B}$, which parametrize a probabilistic hierarchy, while the edge weights are given by the dataset and used to compute the node and edge distribution. We show in Sec. 4.2 that the optimal values of the expected scores share the same intuitive meaning as their discrete counterparts. While the probabilities used in the FPH computation are consistent, their relaxed scores are not consistent with the expected scores under the tree-sampling procedure. In Fig. 2 we show a simple case where Soft-Das does not align with the global optimal value, whereas Exp-Das does.

## 4.2 THEORETICAL ANALYSIS OF EPH AND FPH

The main motivation to use the expected metrics is the property that their global *optimal value*, i.e., the *score* obtained by the globally optimal hierarchy (the *optimizer*), is equal to their discrete counterparts, as we show in Prop. 4.1.

**Proposition 4.1.** *Let $\boldsymbol{A}$ and $\boldsymbol{B}$ be probabilistic transition matrices. Then the following equalities hold,*

$$\min_{\boldsymbol{A}, \boldsymbol{B}} \mathbb{E}_{\hat{\mathcal{T}} \sim P_{\boldsymbol{A}, \boldsymbol{B}}(\mathcal{T})} \left[ \text{Das}(\hat{\mathcal{T}}) \right] = \min_{\hat{\mathcal{T}}} \text{Das}(\hat{\mathcal{T}}) \quad \text{and} \quad \max_{\boldsymbol{A}, \boldsymbol{B}} \mathbb{E}_{\hat{\mathcal{T}} \sim P_{\boldsymbol{A}, \boldsymbol{B}}(\mathcal{T})} \left[ \text{TSD}(\hat{\mathcal{T}}) \right] = \max_{\hat{\mathcal{T}}} \text{TSD}(\hat{\mathcal{T}}) \tag{5}$$

*(See proof in App. A.5)*

Consequently, optimizing our cost function aims to find the optimal discrete hierarchy. Furthermore, we prove in Prop. 4.2 that Soft-Das is a lower bound of Exp-Das, therefore its minimum is a lower bound of the optimal discrete Dasgupta cost.

**Proposition 4.2.** *Let $A$ and $B$ be transition matrices describing a probabilistic hierarchy. Then, Soft-Das will be lower than or equal to the expected Dasgupta cost under the tree-sampling procedure, i.e.,*

$$\text{Soft-Das}(\mathcal{T}) \leq \mathbb{E}_{\hat{\mathcal{T}} \sim P_{\boldsymbol{A}, \boldsymbol{B}}(\mathcal{T})} \left[ \text{Das}(\hat{\mathcal{T}}) \right]. \tag{6}$$

*(See proof in App. A.4)*

In Fig. 2 we provide a specific example illustrating a case where the minimizer of Soft-Das is continuous, and FPH fails to find the optimal hierarchy, i.e., obtaining a minimal Dasgupta cost. We know an integral solution exists for EPH since Exp-Das and Exp-TSD are convex combinations of

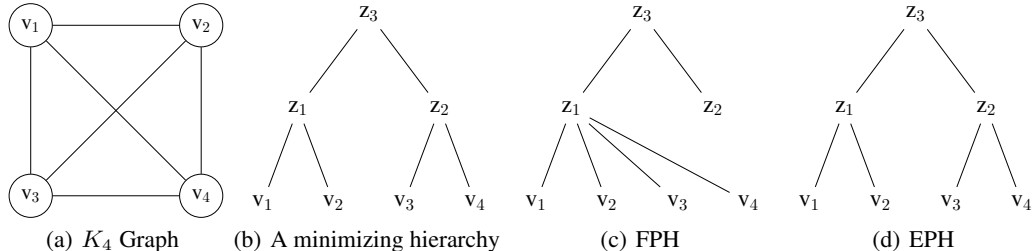

(a) $K_4$ Graph     (b) A minimizing hierarchy     (c) FPH     (d) EPH

Figure 2: An example where FPH fails to infer a minimizing hierarchy. A hierarchy minimizing the Dasgupta cost and the inferred hierarchies by FPH and EPH on the unweighted $K_4$ graph, i.e., every normalized edge weight is equal to $\frac{1}{6}$. While FPH achieves a Dasgupta cost of 4.0 after discretization, the continuous hierarchy has a Soft-Das score below 3.0. On the other hand, EPH finds a minimizing hierarchy with a cost of $\frac{10}{3}$. A weighted example is shown in Fig. 8.

their discrete counterparts. Furthermore, Exp-Das is neither convex nor concave, as we show in App. A.6. In Tab. 1 we provide an overview of properties of the cost functions of FPH and EPH.

Table 1: Properties of Soft-Das, Exp-Das, Soft-TSD, and Exp-TSD.

| Property | Problem Type | Convex/Concave | Integral | Optimal | Consistent |
|---|---|---|---|---|---|
| Soft-Das | Min. | Neither w.r.t. $\boldsymbol{A}$ and $\boldsymbol{B}$ (see Fig.13 (left)) | ✗ | ✗ | ✗ |
| Exp-Das | Min. | Neither w.r.t. $\boldsymbol{A}$ and $\boldsymbol{B}$ (see App.A.6) | ✓ | ✓ | ✓ |
| Soft-TSD | Max. | Convex w.r.t. LCA probabilities (Zügner et al., 2022) | ✓ | ✓ | ✗ |
| Exp-TSD | Max. | - | ✓ | ✓ | ✓ |

### 4.3 UNBIASED COMPUTATION OF EXPECTED SCORES VIA DIFFERENTIABLE SAMPLING

In order to compute the expected scores we can use a closed-form expression. To derive these for Exp-Das and Exp-TSD, we need to be able to calculate the probability $p\left(z = v_i \wedge v_j, z \in \mathrm{anc}(v)\right)$ for which no known solution exists, and the expectation of a logarithm (see Eq. 13 and Eq. 14). An alternative to the closed-form solution is to approximate the expectations via the Monte Carlo method. We propose to approximate Exp-Das and Exp-TSD with $N$ *differentiably* sampled hierarchies $\{\hat{\mathcal{T}}^{(1)}, \ldots, \hat{\mathcal{T}}^{(N)}\}$ (see "Loss computation" in Fig. 1):

$$\text{Exp-Das}(\mathcal{T}) \approx \frac{1}{N} \sum_{i=1}^{N} \text{Das}(\hat{\mathcal{T}}^{(i)}) \quad \text{and} \quad \text{Exp-TSD}(\mathcal{T}) \approx \frac{1}{N} \sum_{i=1}^{N} \text{TSD}(\hat{\mathcal{T}}^{(i)}). \tag{7}$$

However, differentiable sampling of discrete structures like hierarchies is often complex. To this end, our differentiable hierarchy sampling algorithm combines the tree-sampling procedure (Zügner et al., 2022) and the straight-through Gumbel-Softmax estimator (Jang et al., 2016) in three steps: **(1)** We sample the parents of the leaf nodes by interpreting the columns of $\boldsymbol{A}$ as parameters of straight-through Gumbel-Softmax estimators. **(2)** We sample the parents of the internal nodes by interpreting the columns of $\boldsymbol{B}$ as parameters of straight-through Gumbel-Softmax estimators. This procedure is *differentiable* — each step is differentiable — and *expressive* — it can sample any hierarchy with $n$ leaves and $n'$ internal nodes. **(3)** We use the Monte Carlo method to approximate the expectation by computing the arithmetic mean of the scores of the sampled hierarchies. We reuse the *differentiable* computation of Soft-Das and Soft-TSD, which match the discrete scores for discrete hierarchies while providing gradients w.r.t. $\boldsymbol{A}$ and $\boldsymbol{B}$ (see Fig. 1 for an overview).

**Complexity.** Since we sample $N$ hierarchies from $n' + n - 1$ many categorical distributions with $\mathcal{O}(n')$ classes, the sampling process can be done with a complexity of $\mathcal{O}(N \times n \times n' + N \times n'^2)$. The dominating term is the computation of the Das and TSD scores with a complexity of $\mathcal{O}(N \times m \times n'^2)$ for graph datasets and $\mathcal{O}(N \times n^2 \times n'^2)$ for vector datasets (Zügner et al., 2022). This is often efficient as we typically have $n' \ll n$ and for graphs $m \ll n^2$. In Sec. 4.4 we propose a subgraph sampling approach to reduce the complexity to $\mathcal{O}(N \times M \times n'^2 + n^2)$ for large vector datasets, where $M < n^2$.

**Limitations.** While the previously explained MC estimators of the expectations are unbiased in the forward pass, the estimation of the gradients is not (Paulus et al., 2021) and thus impacts the EPH optimization. Furthermore, even though the global optimal values of the expected and discrete scores match, EPH does not guarantee convergence into a global optimum when optimizing using gradient descent methods.

### 4.4 SCALABLE EXP-DAS COMPUTATION VIA SUBGRAPH SAMPLING

As we discussed in the complexity analysis, the limiting factor is $\mathcal{O}(n^2 \times n'^2)$, corresponding to the evaluation of the Dasgupta cost, which becomes prohibitive for large datasets. To reduce the complexity, we propose an *unbiased* subgraph sampling approach. First, we note that the normalized similarities $P(\mathrm{v}_i, \mathrm{v}_j)$ can be interpreted as a probability mass function of a categorical distribution. This interpretation allows the Dasgupta cost to be rewritten as an expectation and approximated via a sampling procedure. More specifically,

$$\mathrm{Das}(\hat{\mathcal{T}}) = \mathbb{E}_{(\mathrm{v}_i, \mathrm{v}_j) \sim P(\mathrm{v}_i, \mathrm{v}_j)} \left[ c(\mathrm{v}_i \wedge \mathrm{v}_j) \right] \approx \frac{1}{M} \sum_{k=1}^{M} c(\mathrm{v}_i^{(k)} \wedge \mathrm{v}_j^{(k)}), \tag{8}$$

where $\{(\mathrm{v}_i^{(1)}, \mathrm{v}_j^{(1)}), \dots, (\mathrm{v}_i^{(M)}, \mathrm{v}_j^{(M)})\}$ are $M$ edges sampled from the edge distribution $P(\mathrm{v}_i, \mathrm{v}_j)$, which can be done in $\mathcal{O}(M + n^2)$ (Kronmal & Peterson, 1979). We refer to this sampling approach as subgraph sampling (see Fig. 1). We can approximate the expected Dasgupta cost using the same procedure. In contrast to Exp-Das, Exp-TSD cannot be easily viewed as an expectation of edges, thus making the approximation via sub-graph sampling impractical. However, since TSD is a metric originally designed for graphs, which are generally sparse, it would yield little benefits.

Note that we end up with two different sampling procedures. First, we have the *differentiable hierarchy sampling* (see Eq. 7). This is necessary to approximate the expectations. Since we do not have a closed-form expression of Exp-Das and Exp-TSD, we sample discrete hierarchies from the probabilistic ones and average the scores. Secondly, we have the *subgraph sampling* (see Eq. 8), which interprets the Dasgupta cost as an expectation. This is done to reduce the computational overhead for vector datasets since the number of pairwise similarities grows quadratically in the number of data points. The estimation is unbiased and introduces an additional parameter, i.e., the number of sampled edges, which allows a trade-off between computational cost and quality. By inserting the probabilistic edge sampling approach into the tree sampling, we estimate Exp-Das to scale it to large vector datasets. An overview of our model is shown in Fig. 1, and a formal description is given in App. B.9.

## 5 EXPERIMENTS

### 5.1 EXPERIMENTAL SETUP

**Datasets.** We evaluate our method on both graph and vector datasets. *Graph datasets:* We use the same graphs and preprocessing as Zügner et al. (2022). More specifically, we use the datasets Polblogs (Adamic & Glance, 2005), Brain (Amunts et al., 2013), Citeseer (Sen et al., 2008), Genes (Cho et al., 2014), Cora-ML (McCallum et al., 2000; Bojchevski & Günnemann, 2018), OpenFlight (Patokallio), WikiPhysics (Aspert et al., 2019), and DBLP (Yang & Leskovec, 2015). To preprocess the graph, we first collect the largest connected component. Secondly, every edge is made bidirectional and unweighted. An overview of the graphs is shown in Tab. 6 in the appendix. *Vector datasets:* We test our method on vector data for the Dasgupta cost. Here we selected the seven datasets Zoo, Iris, Glass, Digits, Segmentation, Spambase, and Letter from the UCI Machine Learning repository (Dua & Graff, 2017). Furthermore, we also use Cifar-100 (Krizhevsky et al., 2009). Digits and Cifar-100 are image datasets, the remaining are vector data. While we only flatten the images of Digits, we preprocess Cifar-100 using the ResNet-101 BiT-M-R101x1 by Kolesnikov et al. (2020), which was pretrained on ImageNet-21k (Deng et al., 2009). More specifically, we use the 2048 dimensional activations of the final layer for each image in Cifar-100 as feature vectors. Furthermore, we normalize all features to have a mean of zero and a standard deviation of one. We compute cosine similarities between all pairs of data points using their normalized features. This results in dense similarity matrices. Finally, we remove the self-loops. Note that in contrast to the graph datasets, the vector data similarities are weighted. An overview is shown in Tab. 7 in the appendix. Since we are in an unsupervised setting, we have no train/test split, i.e. we train and evaluate on the whole graph.

**Baselines.** We compare our model against both discrete and continuous approaches. For discrete approaches, we use the single, average, complete (Hastie et al., 2009), and ward linkage (Ward Jr, 1963) algorithms, respectively referred to as **SL**, **AL**, **CL**, and **WL**. We do not report the results of SL and CL on the graph datasets which do not have edge weights since these methods are not applicable to unweighted graphs. In addition to the linkage algorithms, we also compare to the

Table 2: Results for the graph datasets. Best scores in bold, second best underlined.

| Dataset | Dasgupta cost (↓) | | | | | | | | Tree-sampling divergence (↑) | | | | | | | |
|---|---|---|---|---|---|---|---|---|---|---|---|---|---|---|---|---|
| | PolBl. | Brain | Cites. | Genes | Cora-ML | OpenF. | WikiP. | DBLP | PolBl. | Brain | Cites. | Genes | Cora-ML | OpenF. | WikiP. | DBLP |
| WL | 338.52 | 567.90 | 137.80 | 270.18 | 301.68 | 379.68 | 660.12 | OOM | 26.59 | 25.13 | 62.14 | 60.93 | 52.76 | 50.59 | 42.18 | OOM |
| AL | 355.61 | 556.68 | 83.69 | 196.50 | 292.77 | 363.40 | 658.04 | 36,463 | 25.25 | 28.91 | 67.80 | 66.72 | 55.30 | 52.02 | 43.15 | 38.99 |
| Louv. | 344.47 | 582.45 | 158.79 | 247.27 | 335.57 | 501.29 | 798.75 | 40,726 | 28.86 | 30.74 | 68.09 | 67.51 | 58.18 | 52.97 | 47.01 | 41.40 |
| RSC | 307.70 | 526.17 | 85.41 | 188.82 | 264.62 | 367.36 | 630.53 | OOM | 28.04 | 29.19 | 67.39 | 66.28 | 56.14 | 52.01 | 44.86 | OOM |
| UF | 331.79 | 508.30 | 91.86 | 208.51 | 305.43 | 410.17 | 560.45 | OOM | 21.77 | 24.49 | 60.13 | 59.45 | 48.42 | 47.64 | 42.37 | OOM |
| gHHC | 349.71 | 595.70 | 147.17 | 308.42 | 313.29 | 390.21 | 672.84 | 87,344 | 24.70 | 25.62 | 59.53 | 54.20 | 49.56 | 51.36 | 41.08 | 16.29 |
| HypHC | 272.81 | 519.96 | 416.38 | 632.02 | 594.23 | 529.04 | 678.45 | OOM | 19.65 | 7.26 | 18.98 | 13.00 | 19.18 | 26.82 | 23.92 | OOM |
| FPH | _238.65_ | _425.70_ | _76.03_ | _182.91_ | _257.42_ | _355.61_ | _482.40_ | 31,687 | _31.37_ | _32.75_ | **69.38** | **67.78** | **59.55** | _57.58_ | _49.87_ | _41.62_ |
| EPH | **235.50** | **400.20** | **74.01** | **176.57** | **238.28** | **312.31** | **456.26** | **30,600** | **32.05** | **34.24** | _69.36_ | _67.75_ | _59.41_ | **57.83** | **50.23** | **42.74** |

recursive sparsest cut (**RSC**) (Dasgupta, 2016) and the Louvain method (**Louv.**) (Blondel et al., 2008). For continuous approaches, we use the gradient-based optimization approaches Ultrametric Fitting (**UF**) (Chierchia & Perret, 2019), Hyperbolical Hierarchical Clustering (**HypHC**) (Chami et al., 2020), gradient-based Hyperbolic Hierarchical Clustering (**gHHC**) (Monath et al., 2019), and Flexible Probabilistic Hierarchy (**FPH**) (Zügner et al., 2022). While the linkage algorithms derive a hierarchy based on heuristics or local objectives, UF, HypHC, gHHC, and FPH aim to optimize a relaxed Dasgupta cost or TSD score. For all the methods, we set a time limit of 120 hours and provide a budget of 512GB of memory for each experiment.

**Practical Considerations.** We repeat the randomized methods with five random seeds and report the best score of the discrete hierarchies. We use the same experimental setup as Zügner et al. (2022), i.e., we use $n' = 512$ internal nodes, compress hierarchies using the scheme presented by Charpentier & Bonald (2019), and use DeepWalk embeddings (Perozzi et al., 2014) on the graphs for methods that require features. We train EPH using PAdamax (projected Adamax (Kingma & Ba, 2014)), reduce the learning rate for $B$ by a factor of 0.1 every 1000 epochs, and reset the probabilistic hierarchy to the so-far best discrete hierarchy. To approximate the expectation of EPH, we use 20 samples, except for Spambase, Letter, and Cifar-100, where we use 10, 1, and 1, respectively, to reduce the runtime. On the datasets Digits, Segmentation, Spambase, Letter, and Cifar-100, we train EPH and FPH by sampling $n\sqrt{n}$ edges. On the remaining datasets, we use the whole graph. Both EPH and FPH are initialized using the average linkage algorithm. We train FPH with its original setting and our proposed scheduler and report the minimum of both for each dataset. For the remaining methods, we use the recommended hyperparameters. An overview of the hyperparameters is shown in Tab. 9, and an ablation study in App. B.6. Finally, all methods are evaluated on discrete hierarchies. Instead of sampling a discrete hierarchy from the probabilistic matrices $A$ and $B$, we take the most likely edge for each node as Zügner et al. (2022) proposed. This can be efficiently done by selecting the entry with the highest probability in each row, resulting in the discrete matrices $\hat{A}$ and $\hat{B}$. This approach serves a greedy approximation to extract the most-likely hierarchy from the probabilistic one.

## 5.2 RESULTS

**Graph Dataset Results.** We report the discrete Dasgupta costs and TSD scores for the graph datasets in Tab. 2. EPH achieves 13/16 best and second-best scores otherwise. In particular, EPH, which optimizes Exp-Das, consistently achieves a better Dasgupta cost than FPH, which optimizes Soft-Das. This observation aligns with the theoretical advantages of Exp-Das compared to Soft-Das (see Sec. 4.2). EPH and FPH, which both use the tree-sampling probabilistic framework, always achieve the best results. This highlights the benefit of the tree sampling probabilistic framework for hierarchical clustering. The discrete approaches which use heuristics achieve competitive results but are constantly inferior to EPH.

Furthermore, we can observe that the performance of the linkage algorithms, WL and AL, and the Louvain method is competitive, even though they use heuristics or local objectives to infer a hierarchy. Finally, the inferior performance of gHHC and HypHC can intuitively be explained by the fact that these methods are originally designed for vector datasets. WL, UF, and HypHC were not able to scale to the DBLP datasets within the memory budget. Indeed, they require to compute a dense $n^2$ similarity matrix leading to out-of-memory (OOM) issues.

**Vector Dataset Results.** We report the discrete Dasgupta costs of several methods on the vector datasets in Tab. 3. Similarly to the graph datasets, EPH outperforms all baselines and achieves 7/8 best scores. These results demonstrate the capacity of EPH to also adapt to vector datasets.

Table 3: Dasgupta costs ($\downarrow$) for the vector datasets. Best scores in bold, second best underlined.

| Dataset | Zoo | Iris | Glass | Digits | Segm. | Spam. | Letter | Cifar |
|---|---|---|---|---|---|---|---|---|
| WL | 56.28 | 69.98 | 122.16 | 1126.77 | 1266.17 | 2962.62 | 12241 | 32979 |
| AL | 56.31 | 69.48 | 121.64 | 1121.68 | 1258.22 | 2952.21 | 12181 | 32972 |
| SL | 57.67 | 70.71 | 126.33 | 1166.05 | 1368.21 | 3042.28 | 13166 | 33314 |
| CL | 55.78 | 70.10 | 123.02 | 1140.26 | 1277.48 | 2971.59 | 12396 | 33131 |
| Louv. | 56.26 | 72.31 | 125.94 | 1126.48 | 1238.35 | 2916.48 | 11946 | 32940 |
| RSC | 55.94 | 69.10 | 121.45 | 1119.43 | 1237.20 | 2917.07 | 11895 | 32907 |
| UF | 56.28 | 69.40 | 122.43 | 1137.53 | 1322.86 | 2998.17 | 13090 | OOM |
| gHHC | 60.09 | 69.63 | 123.33 | 1119.74 | 1269.28 | 3018.44 | 12151 | 33089 |
| HypHC | 56.05 | 69.22 | 121.52 | 1118.08 | 1233.07 | 2921.38 | 11930 | OOM |
| FPH | 56.13 | 69.13 | 122.00 | 1132.84 | 1238.45 | 2933.56 | 12197 | 33224 |
| EPH | **55.77** | **69.10** | **120.94** | **1117.58** | **1230.60** | **2916.17** | **11894** | 32913 |

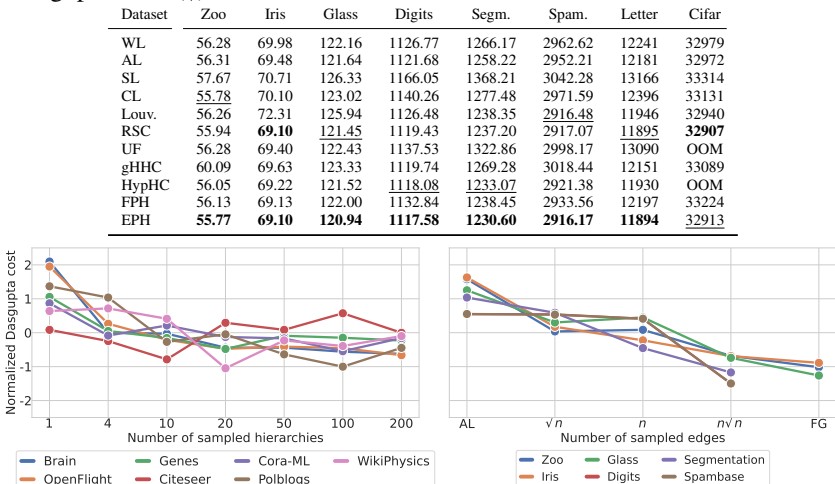

Figure 3: Hyperparameter study. Normalized Dasgupta costs for different numbers of sampled hierarchies (left) and different number of sampled edges (right) after the EPH training, including the average linkage algorithm (AL) and a training on the full graph (FG). The scores are normalized such that each dataset has a mean of zero and a standard deviation of one.

Further, EPH constantly outperforms FPH again. This emphasized the benefit of optimizing expected scores compared to soft scores. In contrast with graph datasets, HypHC performs competitively on vector datasets. This is reasonable since this method is originally designed for vector datasets. FPH performs slightly worse than HypHC on most datasets and is only better on Iris.

**Hyperparameter study.** We show in Fig. 3 (left) the effect of the number of sampled hierarchies on the EPH performances. On one hand, we observe that a large number of sampled hierarchies (i.e. $N \geq 20$) generally yields better results than a small number of sampled hierarchies (i.e. $N \leq 10$) except for Citeseer. Intuitively, a higher number of sampled hierarchies should lead to a more accurate expected score approximation. On the other hand, we observe that a very large number for sampled hierarchies (i.e. $N \geq 100$) might not lead to significant improvements while requiring more computational resources. Intuitively, the noise induced by a lower number of sampled hierarchies could be beneficial to escape local optima. In general, we found that 20 samples lead to satisfactory results for all datasets, thus achieving a good trade-off between approximation accuracy, optimization noise, and computational requirements. We show in Fig. 3 (right) the effect of the number of sampled edges on the EPH performances on vector datasets. Using more edges consistently leads to better results. In particular, going from $n$ to $n\sqrt{n}$ shows a significant performance improvement, while going from $n\sqrt{n}$ to $n^2$ yields only minor improvements. Hence, controlling the amount of sampled edges allows us to scale our method to large datasets while maintaining high performance. On the small datasets Zoo, Iris, and Glass, we use the whole graph, while for the other datasets, we sample $n\sqrt{n}$ edges as a trade-off between runtime and quality of the hierarchical clustering.

**External Evaluation.** As the Dasgupta cost and Tree-Sampling Divergence are internal metrics and do not necessarily measure how closely a derived hierarchy aligns with some external ground truth, we complement the unsupervised quantitative evaluation with an external evaluation. Since we typically do not have access to the ground-truth hierarchies in real-world data, we evaluate EPH on synthetic graph datasets with known ground-truth hierarchies. Furthermore, we analyze to what extent the inferred hierarchies preserve the flat class labels for the vector datasets in Sec. B.5 in the appendix.

We augment the graph datasets with two hierarchical stochastic block models (**HSBMs**), which allow us to compare the inferred hierarchy with the ground truth. As the HSBM graphs are generated in a stochastic process, the ground-truth hierarchy is not necessarily optimal in terms of the Dasgupta cost or Tree-Sampling Divergence. Furthermore, we compute the normalized mutual information (**NMI**) across the different levels of the hierarchies, and the cophenetic correlations for the shortest path distance (SPD) and the Euclidean distance of DeepWalk embeddings (see Tab. 4). We observe that EPH recovers the first three levels of the ground-truth hierarchy almost perfectly. Moreover, the inferred hierarchies by EPH obtain even better scores than the ground-truth hierarchies on the

Table 4: Results of EPH for the HSBMs with $n'$=# Cluster.

| Method | Dasgupta cost ($\downarrow$) | | Tree-sampling divergence ($\uparrow$) | |
|---|---|---|---|---|
| | HSBM Small | HSBM Large | HSBM Small | HSBM Large |
| GT | 26.29 | 130.16 | 43.14 | 51.50 |
| EPH | 26.19 | 121.08 | 43.56 | 51.53 |

| Level | Normalized Mutual Information | | Normalized Mutual Information | |
|---|---|---|---|---|
| Level 1 | 1.0 | 1.0 | 1.0 | 1.0 |
| Level 2 | 1.0 | 1.0 | 1.0 | 1.0 |
| Level 3 | 0.77 | 0.81 | 0.87 | 0.99 |

| | Cophenetic Correlation ($\uparrow$) | | | |
|---|---|---|---|---|
| | Shortest path distance | | DeepWalk Distance | |
| Method | HSBM Small | HSBM Large | HSBM Small | HSBM Large |
| GT | **0.77453** | **0.67839** | 0.94001 | 0.90920 |
| Exp-Das | 0.77174 | 0.67692 | 0.93973 | 0.89788 |
| Exp-TSD | 0.77440 | 0.67838 | **0.94027** | **0.90922** |

(a) Small HSBM - GT     (b) Small HSBM - Exp-Das     (c) Small HSBM - Exp-TSD

(d) Large HSBM - GT     (e) Large HSBM - Exp-Das     (f) Large HSBM - Exp-TSD

Figure 4: Ground truth clusters and dendrograms compared to the inferred ones for the HSBMs.

HSBMs, underlining the remarkable capacity of EPH to optimize the Dasgupta and TSD scores. Furthermore, the TSD objective appears to be a more suitable metric to recover the ground-truth levels of the HSBM in terms of NMI. The cophenetic correlations observed between both Exp-Das and Exp-TSD, are notably high and very close to the GT, with Exp-TSD surpassing the GT on the DeepWalk distances. We attribute this to the fact that a minimal Dasgupta cost favors binary branches, which does not reflect the hierarchies of the HSBMs. Our visual analysis of the graphs and hierarchies (see Fig. 4) confirms this observation. Additional results for FPH are shown in App. B.3.

**Qualitative Evaluation.** We visualize the largest cluster inferred on Cifar-100 using EPH. More specifically, we select the internal nodes with the most directly connected leaves. Furthermore, we sort the images by their probability, i.e., their entry in the matrix $A$. We show the 16 images with the highest probability and the 16 with the lowest probability for the largest cluster in Fig. 5. We observe that the images with high probabilities are all similar and related to insects. This shows that EPH is able to group similar images together. In contrast, the last images with the lowest probability do

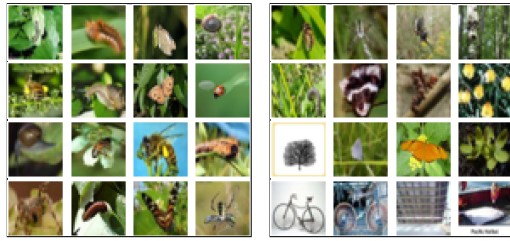

(a) Highest Probability     (b) Lowest Probability

Figure 5: Largest derived cluster on Cifar-100.

not fit into the group. This demonstrates the capacity of EPH to measure the uncertainty in the cluster assignments. We provide additional results with the same behavior for other clusters in App. B.4 (see Fig. 10, Fig. 11, and Fig. 9). Furthermore, we visualize the graph and inferred hierarchies of EPH for OpenFlight in Fig. 12 in the appendix. Both, minimizing Exp-Das and Exp-TSD generate reasonable clusters and are able to successfully distinguish different world regions.

## 6 CONCLUSION

In this work, we propose EPH, a novel end-to-end learnable approach to infer hierarchies in data. EPH operates on probabilistic hierarchies and directly optimizes the expected Dasgupta cost and Tree-Sampling Divergence using *differentiable hierarchy sampling*. We show that the global optima of the expected scores are equal to their discrete counterparts. Furthermore, we present an *unbiased subgraph sampling* approach to scale EPH to large datasets. We demonstrate the capacity of our model by evaluating it on several synthetic and real-world datasets. EPH outperforms traditional and recent state-of-the-art baselines.

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

## A  APPENDIX

### A.1  EQUATIONS OF SOFT-DAS AND SOFT-TSD

In the following, we show the equations of Soft-Das and Soft-TSD.

$$\text{Soft-Das}(\mathcal{T}) = \sum_{\text{v}_i, \text{v}_j \in V} P(\text{v}_i, \text{v}_j) \sum_{\text{z} \in Z} \sum_{v \in V} p\left(\text{z} = \text{v}_i \wedge \text{v}_j\right) p\left(\text{z} \in \text{anc}(\text{v})\right) \tag{9}$$

$$\text{Soft-TSD}(\mathcal{T}) = \sum_{\text{z} \in Z} p(\text{z}) \log \frac{p(\text{z})}{q(\text{z})} \tag{10}$$

$$\text{where } p(\text{z}) = \sum_{\text{v}_i, \text{v}_j \in V} P(\text{v}_i, \text{v}_j) p\left(\text{z} = \text{v}_i \wedge \text{v}_j\right) \tag{11}$$

$$q(\text{z}) = \sum_{\text{v}_i, \text{v}_j \in V} P(\text{v}_i) P(\text{v}_j) p\left(\text{z} = \text{v}_i \wedge \text{v}_j\right) \tag{12}$$

### A.2  CLOSED FORM SOLUTIONS OF EXP-DAS AND EXP-TSD

To compute closed-form solutions of the expectations, the following equations need to be solved:

$$\text{Exp-Das}(\mathcal{T}) = \sum_{\text{v}_i, \text{v}_j \in V} P(\text{v}_i, \text{v}_j) \sum_{\text{z} \in Z} \sum_{v \in V} p\left(\text{z} = \text{v}_i \wedge \text{v}_j, \text{z} \in \text{anc}(\text{v})\right) \tag{13}$$

$$\text{Exp-TSD}(\mathcal{T}) = \sum_{\text{z} \in Z} \mathbb{E}_{\hat{\mathcal{T}} \sim P_{\boldsymbol{A},\boldsymbol{B}}(\mathcal{T} = (\boldsymbol{A}, \boldsymbol{B}))} \left[ p(\text{z}) \log \frac{p(\text{z})}{q(\text{z})} \right]. \tag{14}$$

### A.3  RELATION BETWEEN JOINT AND INDEPENDENT LCA AND ANCESTOR PROBABILISTIES

While the LCA probabilities are crucial to compute Soft-Das, Exp-Das requires the joint LCA and ancestor probabilities, i.e., $p(\text{z}_k = \text{v}_i \wedge \text{v}_j, \text{v} \in \text{anc}(\text{z}_k))$, for the leaves $\text{v}_i$, $\text{v}_j$ and $\text{v}$ and the internal node $\text{z}_k$. In Prop. A.1, we show that the joint probabilities are an upper bound of the product of the single terms.

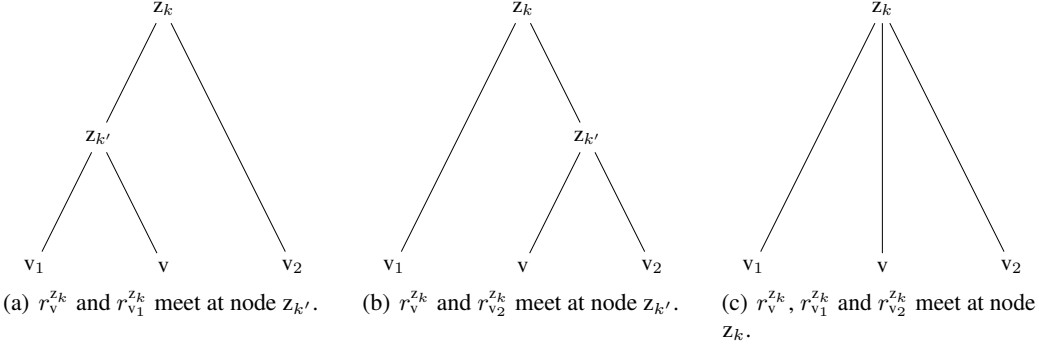

(a) $r_\text{v}^{\text{z}_k}$ and $r_{\text{v}_1}^{\text{z}_k}$ meet at node $\text{z}_{k'}$.  (b) $r_\text{v}^{\text{z}_k}$ and $r_{\text{v}_2}^{\text{z}_k}$ meet at node $\text{z}_{k'}$.  (c) $r_\text{v}^{\text{z}_k}$, $r_{\text{v}_1}^{\text{z}_k}$ and $r_{\text{v}_2}^{\text{z}_k}$ meet at node $\text{z}_k$.

Figure 6: The different cases of the event $p\left(\text{z}_k = \text{v}_1 \wedge \text{v}_2 | \text{z}_k \in \text{anc}(\text{v})\right)$. While the LCA of $\text{v}_1$ and $\text{v}_2$ is $\text{z}_k$ in every case, the LCA of $\text{v}_1$ and $\text{v}$ and the LCA of $\text{v}_2$ and $\text{v}$ are different. We have three cases: either the paths from $\text{v}_1$ or $\text{v}_2$ and $\text{v}$ meet before $\text{z}_k$ at node $\text{z}_{k'}$ (shown in (a) and (b)), or all paths meet for the first time at $\text{z}_k$ (shown in fig. (c)).

**Proposition A.1.** *Let $p$ describe the probability under the tree-sampling procedure, $\text{z}_k$ an internal node, $\text{v}_1$, $\text{v}_2$, and $\text{v}$ leaves. Then, the following inequality holds:*

$$p(\text{z}_k = \text{v}_1 \wedge \text{v}_2) p(\text{z}_k \in \text{anc}(\text{v})) \leq p\left(\text{z}_k = \text{v}_1 \wedge \text{v}_2, \text{z}_k \in \text{anc}(\text{v})\right) \tag{15}$$

*Proof.* First, we observe that the right-hand side of the inequality can be rewritten as:

$$p\left(\text{z}_k = \text{v}_1 \wedge \text{v}_2, \text{z}_k \in \text{anc}(\text{v})\right) = p\left(\text{z}_k = \text{v}_1 \wedge \text{v}_2 | \text{z}_k \in \text{anc}(\text{v})\right) p\left(\text{z}_k \in \text{anc}(\text{v})\right). \tag{16}$$

To prove the non-trivial case $p(z_k \in \text{anc}(v)) \neq 0$, we need to show that the following holds:

$$p(z_k = v_1 \wedge v_2) \leq p(z_k = v_1 \wedge v_2 | z_k \in \text{anc}(v)). \tag{17}$$

Let $r_{v_i}^{z_j} = (v_i, \ldots, z_j)$ denote a path from a leaf $v_i$ to an internal node $z_j$ and let $z_{n'}$ be the root node. Recalling from Zügner et al. (2022) that the paired path probability under the tree-sampling procedure is

$$p((r_{v_1}^{z_{n'}}, r_{v_2}^{z_{n'}})) = p(r_{v_1}^{z_k})p(r_{v_2}^{z_k})p(r_{z_k}^{z_{n'}}), \tag{18}$$

with $z_k = v_1 \wedge v_2$, we can rewrite the LCA probabilities as

$$p(z_k = v_1 \wedge v_2) = \sum_{(r_{v_1}^{z_k}, r_{v_2}^{z_k}):z_k=v_1 \wedge v_2} p(r_{v_1}^{z_k})p(r_{v_2}^{z_k}). \tag{19}$$

Adding the condition $z_k \in \text{anc}(v)$ means there exists a path from the leaf $v$ to the internal node $z_k$. There are three different cases: first, the path meets $r_{v_1}^{z_k}$ and $r_{v_2}^{z_k}$ at $z_k$ for the first time, or the path meets the path $r_{v_1}^{z_k}$ or $r_{v_2}^{z_k}$ in a lower node $z_{k'}$, with $k' < k$. The cases are shown in Fig. 6. In the first case, all three paths are independent. Thus, the LCA probabilities do not change. In the other two cases, they are only independent up to the node $z_{k'}$. The probability for the path $r_{z_{k'}}^{z_k}$ is equal to 1 since we know that $z_k \in \text{anc}(v)$. More formally, the conditional probability is

$$p(z_k = v_1 \wedge v_2 | z_k \in \text{anc}(v)) = \sum_{(r_{v_1}^{z_k}, r_{v_2}^{z_k}):z_k=v_1 \wedge v_2} p(r_{v_1}^{z_k} | z_k \in \text{anc}(v))p(r_{v_2}^{z_k} | z_k \in \text{anc}(v)). \tag{20}$$

Assuming that the path from $v$ to $z_k$ meets the path from $v_1$ to $z_k$ in the node $z_{k'}$ with $k' \leq k$, we have

$$p(r_{v_1}^{z_k} | z_k \in \text{anc}(v))p(r_{v_2}^{z_k} | z_k \in \text{anc}(v)) = p(r_{v_1}^{z_k'})p(r_{v_2}^{z_k}) \geq p(r_{v_1}^{z_k})p(r_{v_2}^{z_k}). \tag{21}$$

The last inequality follows since $r_{v_1}^{z_k'}$ is a subpath of $r_{v_1}^{z_k}$ and therefore has a higher probability. This concludes the proof. □

## A.4 Proof of Proposition 4.2

In the following, we provide the proof of the inequality shown in Prop. 4.2.

*Proof.* To prove it, we first write out the definitions of Soft-Das and the expected Dasgupta cost.

$$\text{Soft-Das}(\mathcal{T}) = \sum_{v_1,v_2} P(v_1, v_2) \sum_z \sum_v P(z = v_1 \wedge v_2)P(z \in \text{anc}(v)) \tag{22}$$

and

$$\mathbb{E}_{\hat{\mathcal{T}} \sim P_{A,B}(\mathcal{T})}\left[\text{Das}(\hat{\mathcal{T}})\right] = \mathbb{E}_{\hat{\mathcal{T}} \sim P_{A,B}(\mathcal{T})}\left[\sum_{v_1,v_2} P(v_1, v_2) \sum_z \sum_v \mathbb{I}_{[z=v_1 \wedge v_2]}\mathbb{I}_{[z \in \text{anc}(v)]}\right] \tag{23}$$

$$= \mathbb{E}_{\hat{\mathcal{T}} \sim P_{A,B}(\mathcal{T})}\left[\sum_{v_1,v_2} P(v_1, v_2) \sum_z \sum_v \mathbb{I}_{[z=v_1 \wedge v_2, z \in \text{anc}(v)]}\right] \tag{24}$$

$$= \sum_{v_1,v_2} P(v_1, v_2) \sum_z \sum_v \mathbb{E}_{\hat{\mathcal{T}} \sim P_{A,B}(\mathcal{T})}\left[\mathbb{I}_{[z=v_1 \wedge v_2, z \in \text{anc}(v)]}\right] \tag{25}$$

$$= \sum_{v_1,v_2} P(v_1, v_2) \sum_z \sum_v P(z = v_1 \wedge v_2, z \in \text{anc}(v)) \tag{26}$$

The proof follows by using Prop. A.1. □

## A.5 Proof of Proposition 4.1

Here we provide the proof of Prop. 4.1.

*Proof.* To prove the left-hand side, we first observe that the expected Dasgupta cost can be rewritten as a convex combination of the Dasgupta costs of all possible hierarchies under the tree-sampling

procedure. More formally,

$$\mathbb{E}_{\hat{\mathcal{T}} \sim P_{\boldsymbol{A},\boldsymbol{B}}(\mathcal{T})} \left[ \text{Das}(\hat{\mathcal{T}}) \right] = \sum_{\hat{\mathcal{T}} \in \mathcal{H}(n,n')} P_{\boldsymbol{A},\boldsymbol{B}}(\hat{\mathcal{T}}) \text{Das}(\hat{\mathcal{T}}) \tag{27}$$

where $\mathcal{H}(n, n')$ describes the set of all valid hierarchies with $n$ leaves and $n'$ internal nodes. Thus, the minimizer of the expected Dasgupta cost is a convex combination of all minimizing hierarchies, with the minimum being equal to the optimal Dasgupta cost. The equation on the right-hand side for TSD can be proved equivalently. □

Note that since the expectation operator is convex, any *discrete optimizer* (i.e., discrete hierarchies achieving the optimum value) of the discrete scores will be an optimizer of the expected scores and vice-versa. In this case, discrete hierarchies are represented by deterministic $\boldsymbol{A}$, $\boldsymbol{B}$ matrices. Only probabilistic hierarchies, which are optimizers of the expected scores, represented by non-discrete $\boldsymbol{A}$ and $\boldsymbol{B}$ matrices, are not optimizers of the discrete scores. This is expected since those probabilistic hierarchies do not belong to the valid input domain of the discrete scores. In addition, any sample we draw from these probabilistic optimizers is also a discrete optimizer of Dasgupta or TSD because of the convexity of the expectation operator.

### A.6 NON-CONVEXITY AND NON-CONCAVITY OF EXP-DAS

Minimizing a convex function using gradient descent is easier than a concave one. Minimizing a concave function heavily depends on the initialization in a constrained setting. Exp-Das($\mathcal{T} = (\boldsymbol{A}, \boldsymbol{B})$) is neither convex nor concave with respect to $\boldsymbol{A}$ and $\boldsymbol{B}$. For both, a counter-example exists. This implies that we can not tell whether Exp-Das converges into a local or global minimum when training. To show that Exp-Das is not concave, it is sufficient to find two hierarchies $\mathcal{T}_1 = (\boldsymbol{A}_1, \boldsymbol{B}_1)$ and $\mathcal{T}_2 = (\boldsymbol{A}_2, \boldsymbol{B}_2)$ such that:

$$\frac{1}{2}\text{Exp-Das}(\mathcal{T}_1) + \frac{1}{2}\text{Exp-Das}(\mathcal{T}_2) \geq \text{Exp-Das}\left(\frac{1}{2}(\mathcal{T}_1 + \mathcal{T}_2)\right), \tag{28}$$

and equivalently to show that it is not convex:

$$\frac{1}{2}\text{Exp-Das}(\mathcal{T}_1) + \frac{1}{2}\text{Exp-Das}(\mathcal{T}_2) \leq \text{Exp-Das}\left(\frac{1}{2}(\mathcal{T}_1 + \mathcal{T}_2)\right), \tag{29}$$

where $\mathcal{T}_1 + \mathcal{T}_2 = (\boldsymbol{A}_1 + \boldsymbol{A}_2, \boldsymbol{B}_1 + \boldsymbol{B}_2)$. In Fig. 7, we show these two examples. In (a) and (b), we show two hierarchies, and in (c), a linear interpolation of these two. The graph in (d) satisfies Eq. 28, while the graph in (e) satisfies Eq. 29. We report the Dasgupta costs for all hierarchy and graph combinations in Tab. 5.

Table 5: Dasgupta costs for all combinations of hierarchies and graphs from Fig. 7.

| Hierarchy | $\hat{\mathcal{T}}_1$ | $\hat{\mathcal{T}}_2$ | $\mathcal{T}_I$ |
|---|---|---|---|
| Convex Example | 3.5 | 3.5 | 3.375 |
| Concave Example | 3.0 | 3.0 | 3.25 |

### A.7 WEIGHTED FAILING SOFT-DAS EXAMPLE

In addition to the example in Fig. 2, we show a weighted graph where FPH fails to find the minimizing hierarchy in Fig. 8.

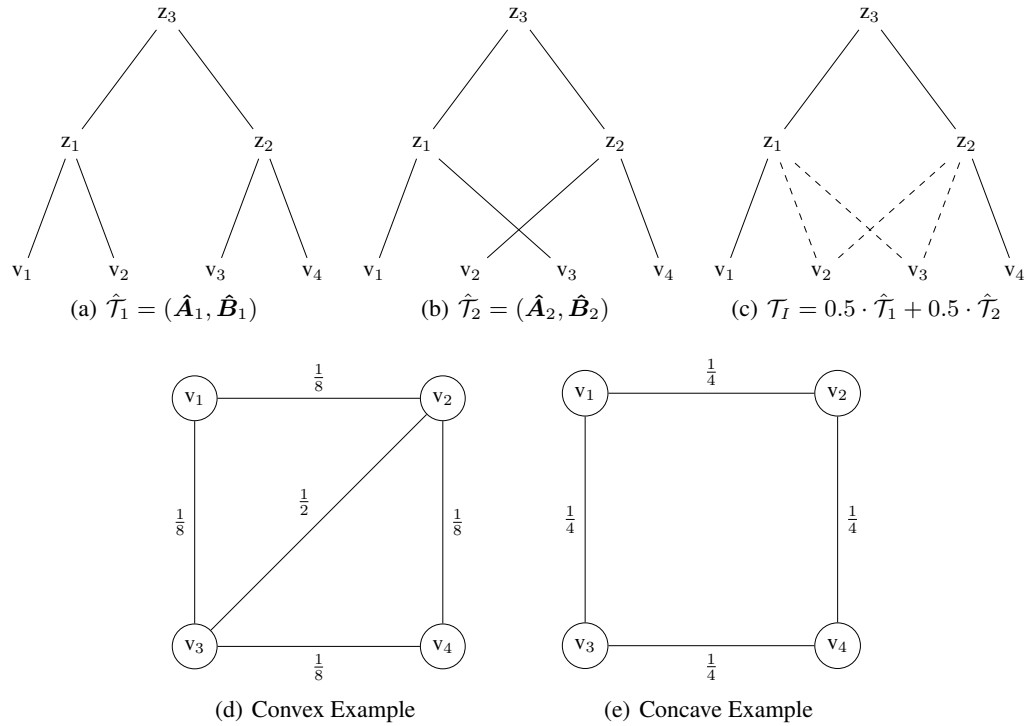

Figure 7: Three hierarchies and two graphs that show that Exp-Das is neither convex nor concave with respect to $\boldsymbol{A}$ and $\boldsymbol{B}$. The hierarchy in (c) is a linear interpolation of the hierarchies in (a) and (b). The graphs in (d) and (e) are counter-examples, with convex and concave behavior, respectively.

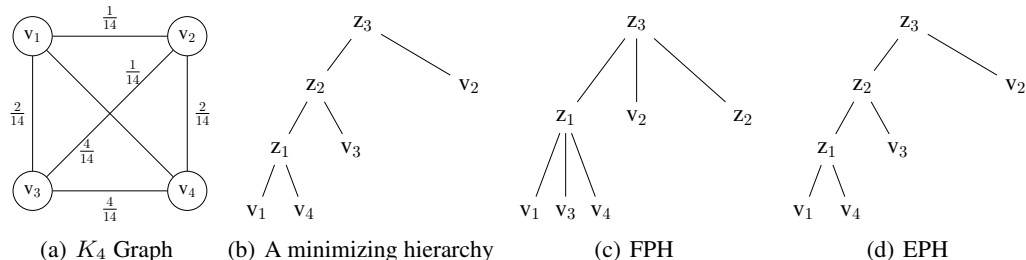

Figure 8: An example of a weighted $K_4$ graph where FPH fails to infer a minimizing hierarchy. While FPH achieves a Dasgupta cost of $\frac{23}{7}$ after discretization, the continuous hierarchy has a Soft-Das score below 3.0. On the other hand, EPH finds a minimizing hierarchy with a cost of 3.0.

# B EXPERIMENTS INFORMATION

## B.1 DATASETS

An overview of the graph and vector datasets is given in Tab. 6 and Tab. 7. The details of the HSBMs are shown in Tab. 8.

Table 6: Overview of the graph datasets.

| Dataset | Number of Nodes | Number of Edges | License |
|---|---|---|---|
| PolBlogs | 1222 | 16715 | n/a |
| Brain | 1770 | 8957 | n/a |
| Citeseer | 2110 | 3694 | n/a |
| Genes | 2194 | 2688 | n/a |
| Cora-ML | 2810 | 7981 | n/a |
| OpenFlight | 3097 | 18193 | OBdL |
| WikiPhysics | 3309 | 31251 | n/a |
| DBLP | 317080 | 1049866 | n/a |

Table 7: Overview of the vector datasets.

| Dataset | Number of Data Points | Number of Attributes | Number of Classes | License |
|---|---|---|---|---|
| Zoo | 101 | 17 | 7 | CC BY 4.0 |
| Iris | 150 | 4 | 3 | CC BY 4.0 |
| Glass | 214 | 10 | 6 | CC BY 4.0 |
| Digits | 1797 | $8 \times 8$ | 10 | CC BY 4.0 |
| Segmentation | 2310 | 19 | 7 | CC BY 4.0 |
| Spambase | 4601 | 57 | 2 | CC BY 4.0 |
| Letter | 20000 | 16 | 26 | CC BY 4.0 |
| Cifar-100 | 50000 | 2048 | 100 | n/a |

Table 8: Overview of the HSBMs.

| Dataset | Number of Nodes | Number of Edges | Number of Clusters |
|---|---|---|---|
| Small HSBM | 101 | 1428 | 15 |
| Large HSBM | 1186 | 27028 | 53 |

## B.2 HYPERPARAMETERS

We show an overview of the hyperparameters we used in Tab. 9. Furthermore, we compare the original FPH results with the tuned results, i.e., the minimum of the original and using the aforementioned scheduler in Tab. 10.

## B.3 HSBM RESULTS FOR FPH

We show the results for FPH on the HSBM graphs in Tab 11.

Table 9: Overview of the Hyperparameters.

| Method | Hyperparameter | Value |
|--------|----------------|-------|
| EPH | LR | Scheduler |
| | Initialization | Average Linkage |
| | Num. Samples | 20 |
| | Num. Samples[*] | 10 |
| | Num. Samples[**] | 1 |
| FPH | LR | $\min\{\text{Scheduler}, 0.05\}$ |
| | LR | $\min\{\text{Scheduler}, 150\}$ |
| | Initialization | Average Linkage |
| | Epochs | 1000 |
| HypHC | LR | $\min\{1e^{-3}, 5e^{-4}, 1e^{-4}\}$ |
| | Temp. | $\min\{1e^{-1}, 5e^{-2}, 1e^{-2}\}$ |
| | LR[***] | $1e^{-3}$ |
| | Temp.[***] | $1e^{-1}$ |
| | Epochs | 50 |
| | Num. Triplets | $n^2$ |
| UF | Loss | $\min\{\text{Dasgupta, Closest+Size}\}$ |
| | LR | 0.1 |
| | Epochs | 500 |
| Scheduler | $LR_A$ (Exp-Das) | 0.1 |
| | $LR_A$ (Exp-Das)[****] | 0.05 |
| | $LR_B$ (Exp-Das) | 0.1 |
| | $LR_B$ (Exp-Das)[*****] | 0.01 |
| | $LR_A$ (Exp-TSD) | 150 |
| | $LR_B$ (Exp-TSD) | 500 |
| | Sampling frequency | 1000 |
| | Sampling frequency[*****] | 2000 |
| | Epochs (Exp-Das) | 10000 |
| | Epochs (Exp-TSD) | 3000 |
| DeepWalk | Embedding Dim. | 10 |
| | Embedding Dim.[*****] | 32 |

[*] Used for DBLP and Spambase
[**] Used for Letter and Cifar-100
[***] Used for Letter
[****] Used for Cifar-100
[*****] Used for DBLP

Table 10: Dasgupta costs of the original FPH (orig.) and with our modifications (tuned). Improvements are highlighted in bold.

| | Dasgupta cost | | | | | | | | Tree-sampling divergence | | | | | | | |
|---|---|---|---|---|---|---|---|---|---|---|---|---|---|---|---|---|
| Dataset | PolBl. | Brain | Cites. | Genes | Cora-ML | OpenF. | WikiP. | DBLP | PolBl. | Brain | Cites. | Genes | Cora-ML | OpenF. | WikiP. | DBLP |
| FPH (orig.) | 262.48 | 503.67 | 77.16 | 183.63 | 257.42 | 355.61 | 537.95 | 31,687 | 31.37 | 32.23 | 69.37 | 67.69 | 58.02 | 57.58 | 49.87 | 41.62 |
| FPH (tuned) | **238.65** | **425.70** | **76.03** | **182.91** | 257.42 | 355.61 | **482.40** | 31,687 | **31.41** | **32.75** | **69.38** | **67.78** | **59.55** | 57.58 | 49.87 | 41.62 |

Table 11: Results of FPH for the HSBMs with $n'$=# Cluster.

|  | Dasgupta cost | | Tree-sampling divergence | |
| --- | --- | --- | --- | --- |
| Method | HSBM Small | HSBM Large | HSBM Small | HSBM Large |
| GT | 26.29 | 130.16 | 43.14 | 51.50 |
| FPH | 27.84 | 127.99 | 43.53 | 51.53 |
| Level 1 | 1.0 | 0.99 | 1.0 | 1.0 |
| Level 2 | 1.0 | 0.95 | 1.0 | 1.0 |
| Level 3 | 0.77 | 0.81 | 0.87 | 0.99 |

## B.4 ADDITIONAL VISUALISATIONS

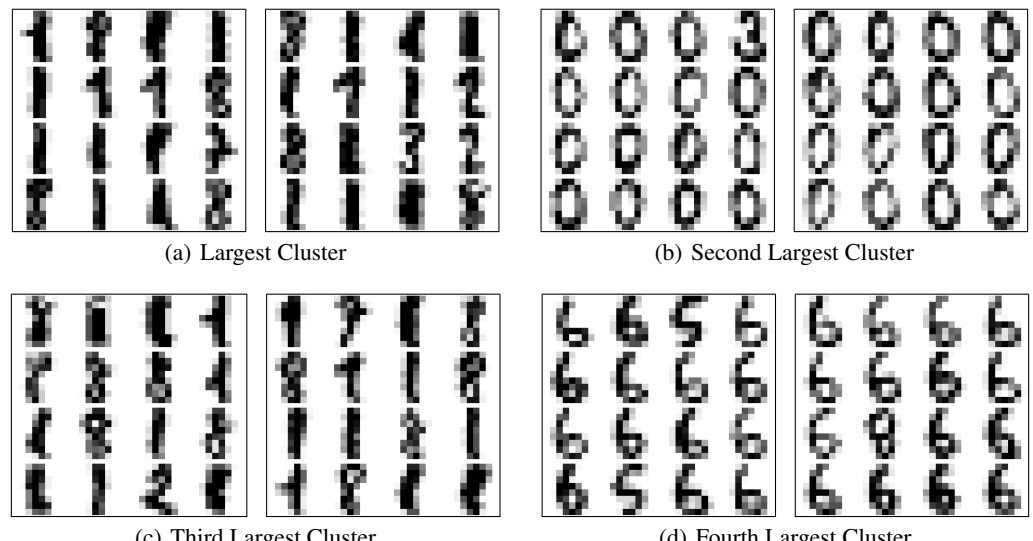

(a) Largest Cluster                    (b) Second Largest Cluster

(c) Third Largest Cluster              (d) Fourth Largest Cluster

Figure 9: Largest derived clusters on Digits. On the left in each subplot the 16 images with the highest probability, on the right the 16 images with the lowest probability.

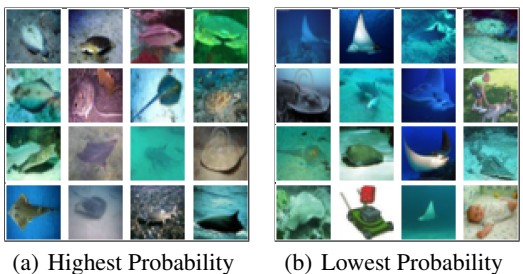

(a) Highest Probability       (b) Lowest Probability

Figure 10: Second largest derived cluster on Cifar-100.

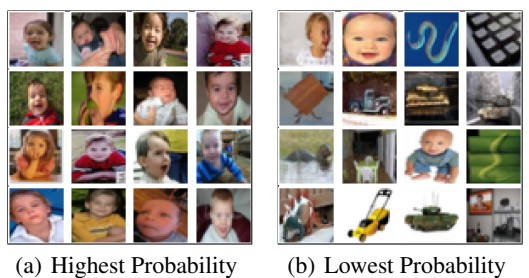

(a) Highest Probability          (b) Lowest Probability

Figure 11: Third largest derived cluster on Cifar-100..

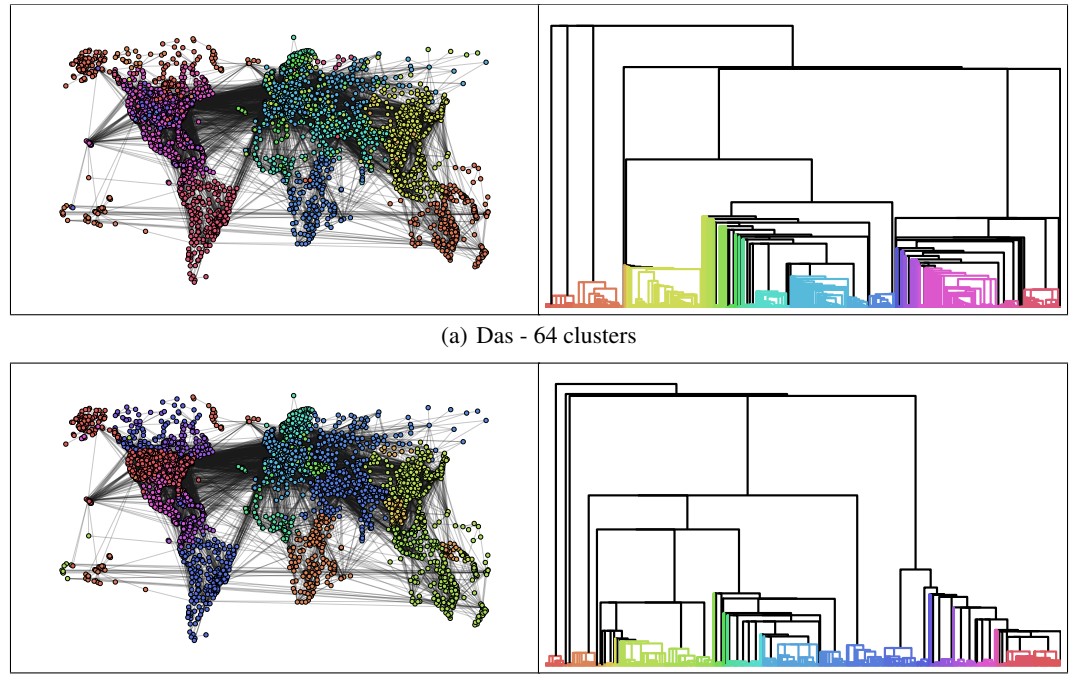

(a) Das - 64 clusters

(b) TSD - 64 clusters

Figure 12: Inferred clusters inferred by EPH optimizing Exp-Das and Exp-TSD. 64 clusters are highlighted in the graphs and dendrograms.

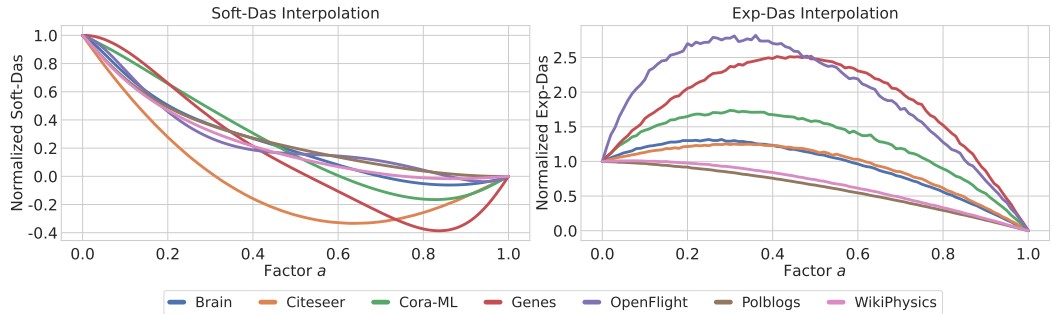

Figure 13: Linear interpolation of Soft-Das and Exp-Das scores from the average linkage hierarchy to the hierarchy inferred by Exp-Das.

Table 12: Accuracies for the vector datasets. Best scores in bold, second best underlined.

| Dataset | Zoo | Iris | Glass | Digits | Segm. | Spam. | Letter | Cifar |
|---------|------|------|-------|--------|-------|-------|--------|-------|
| WL | 0.74 | 0.76 | _0.44_ | **0.82** | **0.61** | 0.76 | **0.33** | **0.49** |
| AL | 0.80 | _0.82_ | 0.43 | 0.65 | 0.48 | _0.82_ | _0.31_ | 0.16 |
| SL | **0.84** | 0.67 | 0.37 | 0.10 | 0.15 | 0.61 | 0.04 | 0.01 |
| CL | _0.81_ | 0.77 | 0.40 | 0.59 | 0.51 | 0.74 | 0.26 | _0.33_ |
| Louv. | 0.60 | **0.83** | 0.42 | _0.68_ | 0.29 | **0.86** | 0.18 | 0.05 |
| RSC | 0.41 | 0.35 | 0.38 | 0.39 | 0.51 | 0.84 | 0.11 | 0.05 |
| UF | 0.60 | 0.55 | 0.43 | 0.43 | _0.54_ | 0.54 | 0.04 | OOM |
| gHHC | 0.59 | 0.71 | 0.42 | 0.51 | 0.30 | 0.69 | 0.08 | 0.01 |
| HypHC | 0.79 | _0.82_ | **0.52** | 0.42 | 0.29 | 0.60 | 0.10 | OOM |
| FPH | 0.58 | **0.83** | 0.40 | 0.20 | 0.44 | 0.61 | 0.06 | 0.03 |
| EPH | 0.70 | **0.83** | 0.40 | 0.65 | 0.53 | **0.86** | 0.14 | 0.18 |

## B.5 ADDITIONAL EXTERNAL EVALUATION

Additionally to the unsupervised metrics, we investigate whether the inferred hierarchies on the vector datasets preserve the flat ground truth class labels. To do this, we flatten the derived hierarchies and compare inferred clusters with the available ground-truth labels by applying the Hungarian algorithm to align the cluster assignments with the labels, as explained by Zhou et al. (2022). Using this procedure, we compute the accuracies, which we show in Tab. 12. While the linkage algorithms were inferior to the continuous optimization algorithms in terms of Dasgupta cost, they dominate here. EPH, trained on Exp-Das, yields the best accuracies only on Iris and Spambase. As the linkage algorithms and Louvain generate hierarchies using heuristics while the continuous methods aim to minimize the Dasgupta cost the results are not surprising, since the Dasgupta cost and other metrics do not necessarily go hand in hand.

## B.6 ABLATION

For our ablation study, we use a simplified optimization scheme. More specifically, we use a fixed learning rate of 0.05 and only train for 1000 epochs.

**Constrained vs. Unconstrained Optimization.** We require the rows of the matrices $A$ and $B$ to be row-stochastic. There are several possibilities to enforce this. Either we can perform constrained optimization using projections onto the probabilistic simplex or simply perform a softmax operation over the rows. In Tab. 13, we show a comparison of the Dasgupta costs on the graph datasets for several graph datasets. We can observe that the constrained optimization,i.e., using projections

Table 13: Dasgupta costs for constrained and unconstrained optimization on several graph datasets with $n' = 512$ internal nodes.

| Dataset | PolBlogs | Brain | Citeseer | Genes | Cora-ML | OpenFlight | WikiPhysics |
|---------|----------|-------|----------|-------|---------|-----------|-------------|
| Constrained | **252.55** | **428.40** | **74.84** | **178.90** | **242.38** | **324.45** | **481.92** |
| Unconstrained | 272.60 | 457.62 | 79.47 | 188.02 | 269.10 | 349.20 | 526.99 |

after each step yields better results than the unconstrained optimization on every graph. This aligns with the findings of Zügner et al. (2022). Therefore, we recommend using constrained optimization.

**Initialization.** The initialization of a model can play a crucial role. Zügner et al. (2022) found that using the AL algorithm as initialization yields substantial improvements. Therefore, we compare both initializations and additionally test using their algorithm FPH as initialization. We show the Dasgupta costs for several graph datasets in Tab. 14. As expected, using the AL algorithm or FPH

Table 14: Dasgupta costs for different initializations on several graph datasets with $n' = 512$ internal nodes. In the first three rows the initial Dasgupta costs and in the last three rows the Dasgupta costs after the training. Best scores in bold, second best underlined.

| Dataset | PolBlogs | Brain | Citeseer | Genes | Cora-ML | OpenFlight | WikiPhysics |
|---|---|---|---|---|---|---|---|
| Random | 914.11 | 1285.68 | 1574.76 | 1621.82 | 2107.65 | 2302.13 | 2479.51 |
| AL | 355.60 | 556.68 | 83.69 | 196.50 | 292.77 | 363.40 | 658.04 |
| FPH | 262.47 | 453.17 | 77.05 | 179.55 | 257.42 | 355.61 | 538.47 |
| Exp-Das (Random) | 275.44 | 499.12 | 307.52 | 368.66 | 564.35 | 502.22 | 624.27 |
| Exp-Das (AL) | 252.55 | **428.40** | **74.84** | 178.90 | **242.87** | **324.45** | **481.92** |
| Exp-Das (FPH) | **251.55** | 431.15 | 74.88 | **177.32** | 245.79 | 326.46 | 527.02 |

as initialization yields significant improvements over a random initialization. Even though the FPH initialization starts with a better hierarchy, the resulting hierarchies are inferior to the AL initialization. This could be caused by local minima, in which the model gets stuck. We recommend using AL as initialization since it performs best on most datasets and has a lower computational cost than FPH.

**Direct vs. Embedding Parametrization.** Additionally to the direct parametrization of the matrices $A$ and $B$, we test an embedding parametrization for each node in the hierarchy. More specifically, we use $d$-dimensional embeddings for the leaves and internal nodes. We perform a softmax operation with an additional learnable temperature parameter $t_i$ to infer $A$ and $B$ over the cosine similarities between the embeddings. The main advantage of the embedding approach is that additionally to the hierarchical clustering, we gain node embeddings that can be used for downstream tasks such as classification or regression. We test the embedding parametrization with $d = 128$ on several graph datasets. Once we let $t_i$ be learnable, and once we freeze it to $t_i = 1$. We compare the results to the constrained optimization. While we train the direct parametrization for 1000 epochs, the embedding approach is trained for 20000 epochs. This is done to ensure convergence since it is randomly initialized. We show the results in Tab. 15. First, we observe that not using a temperature parameter yields substantially

Table 15: Dasgupta costs for the direct and embedding parametrization on several graph datasets with $n' = 512$ internal nodes. Best scores in bold, second best underlined.

| Dataset | PolBlogs | Brain | Citeseer | Genes | Cora-ML | OpenFlight | WikiPhysics |
|---|---|---|---|---|---|---|---|
| Direct | 252.55 | **428.40** | **74.84** | **178.90** | **242.38** | **324.45** | **481.92** |
| Embedding ($t_i = 1$) | 451.63 | 659.11 | 1008.23 | 1146.86 | 1261.91 | 968.41 | 1108.42 |
| Embedding | **249.84** | 440.29 | 213.99 | 290.19 | 409.68 | 373.07 | 514.91 |

worse results. Furthermore, the embedding parametrization is inferior to the direct parametrization, even though it was trained for 20000 epochs, while the constrained optimization was only trained for 1000. Only on the dataset PolBlogs the embedding approach is slightly better than the direct parametrization. We attribute the inferior performance to the random initialization and the fact that we have to use a softmax operation instead of projections. Our results are in line with the ablation study of Zügner et al. (2022). They also parametrized their model using embeddings and used the softmax function on the negative Euclidean distances to infer the matrices $A$ and $B$. Since the embedding approach yields worse results with longer training times, we recommend using the direct parametrization.

**Number of Internal Nodes.** As in many real-world problems we do not know the number of internal nodes $n'$ beforehand in our experiments. While increasing $n'$ generally leads to more refined and expressive hierarchies, it reduces interpretability and comes with a higher computational cost. To select the hyperparameter $n'$, we test various choices on several datasets. We show the corresponding Dasgupta costs and TSD scores in Fig. 14 and Fig. 15. We found that $n' = 512$ is sufficient to capture most information. In practice, we recommend using the Elbow method.

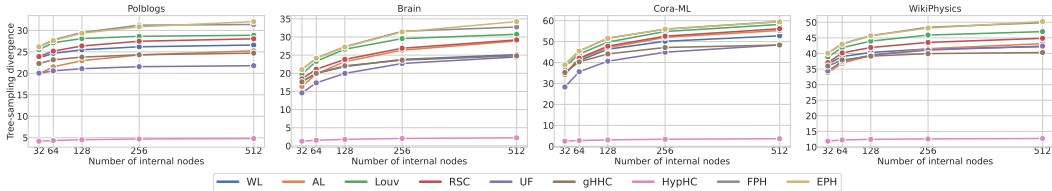

Figure 14: Dasgupta costs for different numbers of internal nodes.

Figure 15: TSD scores for different numbers of internal nodes.

Table 16: Standard Deviations for the graph datasets.

| | Dasgupta cost | | | | | | | | Tree-sampling divergence | | | | | | | |
|---|---|---|---|---|---|---|---|---|---|---|---|---|---|---|---|---|
| Dataset | PolBl. | Brain | Cites. | Genes | Cora-ML | OpenF. | WikiP. | DBLP | PolBl. | Brain | Cites. | Genes | Cora-ML | OpenF. | WikiP. | DBLP |
| gHHC | 7.43 | 16.19 | 7.49 | 12.05 | 192.38 | 200.76 | 42.82 | 4,887.56 | 0.09 | 0.18 | 0.12 | 0.16 | 5.93 | 7.97 | 0.16 | 0.28 |
| HypHC | 6.37 | 9.05 | 22.93 | 16.54 | 51.74 | 28.83 | 36.64 | OOM | 0.47 | 0.64 | 0.71 | 0.35 | 1.47 | 1.02 | 1.22 | OOM |
| EPH | 0.43 | 1.31 | 0.10 | 2.17 | 1.02 | 2.49 | 2.95 | 35.89 | 0.31 | 0.39 | 0.01 | 0.07 | 0.02 | 0.12 | 0.15 | 0.03 |

Table 17: Standard Deviations for the vector datasets.

| | Dasgupta cost | | | | | | | | Accuracy | | | | | | | |
|---|---|---|---|---|---|---|---|---|---|---|---|---|---|---|---|---|
| Dataset | Zoo | Iris | Glass | Digits | Segmentation | Spambase | Letter | Cifar | Zoo | Iris | Glass | Digits | Segmentation | Spambase | Letter | Cifar |
| gHHC | 0.03 | 0.35 | 2.30 | 5.95 | 40.55 | 1.02 | 167.34 | 167.34 | 0.09 | 0.04 | 0.04 | 0.08 | 0.01 | 0.03 | 0.03 | 0.01 |
| HypHC | 0.08 | 0.58 | 0.27 | 1.37 | 0.46 | 2.04 | 45.10 | OOM | 0.05 | 0.01 | 0.05 | 0.07 | 0.06 | <0.01 | 0.01 | OOM |
| FPH | - | - | - | 5.29 | 3.07 | 19.31 | 96.52 | 528.68 | - | - | - | 0.04 | 0.06 | 0.01 | 0.02 | 0.01 |
| EPH | 0.01 | 0.01 | 0.01 | 0.11 | 0.22 | 1.11 | 2.42 | 5.73 | 0.03 | <0.01 | 0.01 | 0.06 | 0.05 | <0.01 | 0.01 | 0.01 |

**Number of Sampled Hierarchies.** Another crucial hyperparameter for EPH is the number of sampled hierarchies. Additionally to Fig. 3, we provide the raw Dasgupta costs and standard errors after the training in Fig. 16. Furthermore, we show the influence of the number of samples to approximate the expected Dasgupta cost on randomly initialized hierarchies in Fig. 17.

## B.7 STANDARD DEVIATIONS

We show the standard deviations of the randomized models on the graph datasets in Tab. 16 and for the vector datasets in Tab. 17.

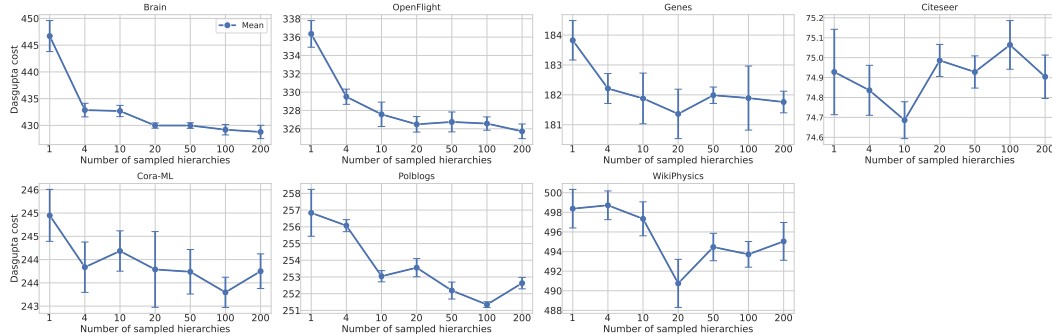

Figure 16: Dasgupta costs and standard error for different numbers of sampled hierarchies after the EPH training.

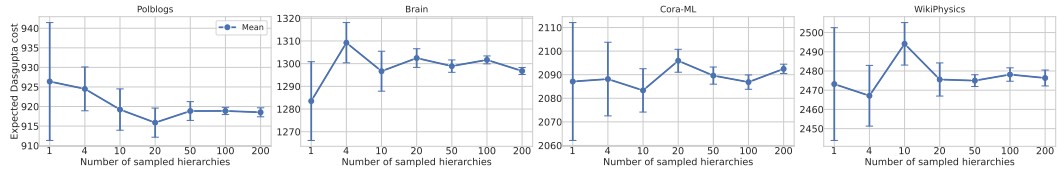

Figure 17: Approximated Expected Dasgupta costs for different numbers of sampled hierarchies for randomly initialized probabilistic hierarchies.

## B.8 RUNTIMES

Table 18: Runtime in seconds for the graph datasets with $n' = 512$ internal nodes.

|  | PolBlogs | Brain | Citeseer | Genes | Cora-ML | OpenFlight | WikiPhysics | DBLP |
|---|---|---|---|---|---|---|---|---|
| # Nodes | 1222 | 1770 | 2110 | 2194 | 2810 | 3097 | 3309 | 317080 |
| # Edges | 16715 | 8957 | 3694 | 2688 | 7981 | 18193 | 31251 | 1049866 |
| WL | 1 | <1 | <1 | <1 | <1 | 1 | 1 | OOM |
| AL | 1 | 1 | <1 | <1 | 1 | 1 | 2 | 101 |
| Louv. | 1 | 1 | 1 | 1 | 1 | 1 | 1 | 1031 |
| RSC | 92 | 78 | 104 | 427 | 280 | 626 | 863 | OOM |
| UF | 9 | 3 | 1 | 1 | 2 | 4 | 6 | OOM |
| gHHC | 75 | 79 | 73 | 78 | 73 | 79 | 83 | 15630 |
| HypHC | 2043 | 4163 | 5981 | 6816 | 11557 | 14278 | 16778 | OOM |
| FPH | 452 | 547 | 345 | 373 | 644 | 592 | 667 | 6647 |
| EPH | 3834 | 3402 | 2609 | 2848 | 3322 | 4419 | 6389 | 224331 |

Table 19: Runtime in seconds for the vector datasets with $n' = \min\{n - 1, 512\}$ internal nodes.

|  | Zoo | Iris | Glass | Digits | Segmentation | Spambase | Letter | Cifar-100 |
|---|---|---|---|---|---|---|---|---|
| # Points | 101 | 150 | 214 | 1797 | 2310 | 4601 | 20000 | 50000 |
| WL | <1 | <1 | <1 | 8 | 13 | 51 | 983 | 8316 |
| AL | <1 | <1 | <1 | 8 | 13 | 51 | 985 | 8564 |
| SL | <1 | <1 | <1 | 8 | 13 | 51 | 975 | 8494 |
| CL | <1 | <1 | <1 | 8 | 13 | 52 | 986 | 8594 |
| Louv. | <1 | <1 | <1 | 8 | 14 | 55 | 1065 | 7324 |
| RSC | 1 | 1 | 2 | 27 | 40 | 127 | 2009 | 14110 |
| UF | <1 | <1 | <1 | 9 | 14 | 57 | 1132 | OOM |
| gHHC | 47 | 57 | 59 | 83 | 66 | 89 | 110 | 8462 |
| HypHC | 47 | 60 | 77 | 3385 | 5814 | 26933 | 250792 | OOM |
| FPH | 87 | 93 | 144 | 2586 | 3963 | 13876 | 134845 | 427557 |
| EPH | 1058 | 1541 | 2407 | 23574 | 44251 | 31435 | 130227 | 430322 |

We report the runtimes for EPH and the baselines in Tab. 18 and Tab. 19. While HypHC, FPH, and EPH are executed on a GPU (NVIDIA A100), the remaining methods do not support or did not

require GPU acceleration. Since gHHC has a lower computational runtime than the other randomized methods, we run it with 50 random seeds instead of 5.

## B.9 PSEUDOCODE

In the following, we provide a formal description of our EPH algorithm, the subgraph sampling, and how we normalize graphs.

---

**Algorithm 1** EPH

---
**Input:** $\mathcal{G} = (V, E)$: Graph
**Input:** $\mathcal{T} = (\boldsymbol{A}, \boldsymbol{B})$: Initial hierarchy
**Input:** $\alpha$: Learning rate
**Input:** $K$: Number of sampled hierarchies
**for** $t = 1, \ldots$ **do**
   $g_t \leftarrow 0$
   **for** $k = 1, \ldots, K$ **do**
      $\hat{\mathcal{G}} \leftarrow \text{SampleSubgraph}(\mathcal{G})$
      $\hat{\mathcal{T}} \sim P_{\boldsymbol{A},\boldsymbol{B}}(\mathcal{T})$
      $g_t \leftarrow g_t + \nabla_{\mathcal{T}} \text{Score}(\hat{\mathcal{G}}, \hat{\mathcal{T}})$
   **end for**
   $\mathcal{T}_t \leftarrow \mathcal{T}_{t-1} - \frac{\alpha}{K} g_t$
   $\mathcal{T}_t \leftarrow P(\mathcal{T}_t)$     //simplex projection
**end for**
**return** $\hat{\mathcal{T}}_t$

---

**Algorithm 2** NormalizeGraph

---
**Input:** $\mathcal{G} = (V, E)$: Graph
$P(\mathrm{v}_i, \mathrm{v}_j) \leftarrow \frac{w_{i,j}}{\sum_{\mathrm{u},\mathrm{v} \in V} w_{u,v}}$
$P(\mathrm{v}_i) \leftarrow \frac{w_i}{\sum_j w_j}$

---

**Algorithm 3** SampleSubgraph

---
**Input:** $\mathcal{G} = (V, E)$: Graph
**Input:** $M$: Number of sampled edges
$\hat{E} \leftarrow \text{MultiSet}()$     //allow duplicate edges
**for** $m = 1 \ldots \mathrm{M}$ **do**
   $e = (\mathrm{v}_i, \mathrm{v}_j) \sim P(\mathrm{v}_i, \mathrm{v}_j)$
   $\hat{E}.\text{add}(e)$
**end for**
$\hat{\mathcal{G}} \leftarrow (V, \hat{E})$
$\text{NormalizeGraph}(\hat{\mathcal{G}})$
**return** $\hat{\mathcal{G}}$

---

