# OpenReview forum: "Expected Probabilistic Hierarchies"
_ICLR.cc/2024/Conference — Submitted to ICLR 2024_

### Official Review · Reviewer_geHj · 2023-10-29

**Soundness:** 3 good
**Presentation:** 2 fair
**Contribution:** 2 fair
**Rating:** 5
**Confidence:** 3

**Summary:**

This paper introduces a differentiable optimization approach for solving the hierarchical clustering problem. Specifically, for the Dasgupta’s objective and TSD objective. To this end, the authors begin by encoding the tree structure as two 0-1 matrices $\hat{A}, \hat{B}$. Then they relax the integer constraint and obtain two continuous matrices $A,B$, which can be interpreted as the probability distribution over the discrete tree structure. Consequently, the goal is transformed into optimizing the expectation over the distribution represented by $A$ and $B$. Moreover, the authors prove that the optimal value of the expectation is equivalent to the discrete version.

In the optimization procedure, the authors replace the expectation with its appriximation, the mean computed from sampled hierarchies. Due to the high complexity of computing Dasgupta’s objectives, they employ subgraph sampling techniques to accelerate the evaluation process.

Finally, the proposed method is extensively evaluated on diverse datasets to assess its performance and effectiveness.

**Strengths:**

- The overall completeness of this work is good. It demonstrates clarity in writing and provides detailed explanations.
- In my opinion, the expected objectives presented in this paper (eq. 4) are more reasonable compared to previous work. For instance, considering $Das(\tilde{T})$, $c(v_i\wedge v_j)$ is the number of leaves under $LCA(v_i, v_j)$ and it should be computed as $\sum_{z\in Z}\sum_{v\in V}\Pr(z=LCA(v_i,v_j)\ and \text{ v is under z})$ in the probabilistic tree. The two conditions in the formula are not independent hence eq. 13 is more accurate than eq. 9.
- The new scaling method proposed by this work has better explainability than its counterpart in [1].
- The experiment results demonstrate that EPH method is competitive in practical application.

[1] End-to-End Learning of Probabilistic Hierarchies on Graphs

**Weaknesses:**

I have not found obvious weaknesses of this paper. However, there is a concern about the novelty contribution when it is compared with [1]. The overal strategy and presentation closely resemble the scheme of [1]. Both the expected objectives and scaling trick appear to be modifications of the results presented in [1].

**Questions:**

- The example $K_4$ graph illustrates the advantage of EPH over FPH. Can you provide an example graph whose edges weights are not the same? As in the context of hierarchical clustering, unweighted graph has no real hierarchy. An instance with non-uniform weights would be more persuasive.

---

> ### Author Response · Authors · 2023-11-17
>
> We thank the reviewer for their valuable feedback. We updated the manuscript based on the feedback with important text modifications highlighted in blue. In the following, we address their comments.
>
>
> **Comment:** Similarity to FPH.
> **Response:** While our work builds upon the work of Zügner et al., it addresses multiple drawbacks of FPH. We prove that Soft-Das is a lower bound of the discrete Dasgupta cost (Sec. 4.2) and present a simple example where it fails to find the minimizing hierarchy. Based on these observations, we propose our cost functions, the expected metrics (Sec. 4.1). We motivate them by showing that the optimal hierarchies align with their discrete counterparts, unlike Soft-Das (Sec. 4.2). Furthermore, we introduce an unbiased subgraph sampling scheme, allowing us to employ both, FPH and EPH, to vector datasets. In an exhaustive empirical evaluation, we demonstrate the superiority of EPH on graph and vector datasets.
>
> **Comment:** Weighted example where FPH fails.
> **Response:** We decided to show the unweighted $K_4$ graph as an example since it is the most simple case. We included another weighted example with four nodes in Appendix A.7. As in the previous case, EPH finds the minimizing hierarchy, while FPH fails to do so.
>
> We hope that we have satisfactorily addressed the reviewer's questions and concerns. We are happy to address any remaining remarks.

---

> ### Comment · Reviewer_geHj · 2023-11-20
>
> Thanks for your response. I have further questions about the subgraph sampling procedure.
>
> The process of sampling edges from the edge distribution $P(v_i, v_j)$ entails an $\Omega(n^2)$ complexity (for vector data). Is this approach less scalable compared to the node-sampling method employed in [1]? Intuitively, the node-sampling technique does not require the preparation of weights for all edges, potentially having a much better scalability.
>
> [1] End-to-End Learning of Probabilistic Hierarchies on Graphs

---

> > ### Author Response · Authors · 2023-11-20
> >
> > > The process of sampling edges from the edge distribution $P(v_i, v_j)$ entails an $\Omega(n^2)$ complexity (for vector data). Is this approach less scalable compared to the node-sampling method employed in [1]? Intuitively, the node-sampling technique does not require the preparation of weights for all edges, potentially having a much better scalability.
> >
> >
> >
> > The preprocessing step, i.e., construction of the alias table, has a linear complexity w.r.t. the pairwise similarities, and therefore quadratic w.r.t. the leaves, i.e., $\mathcal{O}(n^2)$ [1, 2]. This preprocessing step gets amortized during the training. In contrast, the batching scheme employed by FPH does not require this preprocessing step. However, this difference does not affect the overall complexity as both FPH and EPH initialize their hierarchies with the average linkage algorithm, which also has an $\mathcal{O}(n^2)$ complexity. Therefore, both methods have the same scalability.
> >
> > [1] **Vose, Michael D.** "A linear algorithm for generating random numbers with a given distribution." IEEE Transactions on software engineering 17, no. 9 (1991): 972-975.
> >
> > [2] **Wang, Pengyu, Chao Li, Jing Wang, Taolei Wang, Lu Zhang, Jingwen Leng, Quan Chen, and Minyi Guo.** "Skywalker: Efficient alias-method-based graph sampling and random walk on gpus." In 2021 30th International Conference on Parallel Architectures and Compilation Techniques (PACT), pp. 304-317. IEEE, 2021.

---

> > > ### Comment · Reviewer_geHj · 2023-11-23
> > >
> > > Thanks for your clarifications.

---

### Official Review · Reviewer_gKno · 2023-10-31

**Soundness:** 3 good
**Presentation:** 3 good
**Contribution:** 2 fair
**Rating:** 5
**Confidence:** 3

**Summary:**

The authors propose a probabilistic model to learn a hierarchical clustering by optimizing the expectation of a quality metric (TSD or Dasgupta score).

They show that if this criterion has the same optimal value as the discrete counterpart (unlike say a continous relaxation based approach), and so is a reasonable target to optimize for. They use a end-to-end gradient based optimizer to optimize for this target.

Experiments were performed to show that their proposed method outperforms reasonable baselines, including a simple relaxation based method.

**Strengths:**

Use of hierarchical sampling to enable end-to-end differentiable optimization instead of the more obvious relaxation approach is interesting.

**Weaknesses:**

While the resulting clustering does seem to be improved (judged by the improvement in the target criteria), it is unclear how much more expensive this process is compared to the baseline, or how the quality changes with changes in the number of samples used.

**Questions:**

Please include some more details around 1) how much the results vary with the number of samples used 2) speed comparisons

---

> ### Author Response · Authors · 2023-11-17
>
> We want to thank the reviewer for their feedback and questions. We updated the manuscript based on the feedback with important text modifications highlighted in blue. In the following, we address their comments.
>
> **Comment:** How much more expensive is this process compared to the baseline?
> **Response:** We include the runtimes for the experiments in Tab. 18 and Tab. 19 in the Appendix. While EPH is slower than the heuristic-based baselines, it performs similarly to the optimization-based baselines HypHC and FPH. EPH has a shorter runtime than HypHC on the graphs and large vector datasets. Furthermore, it matches the runtime of FPH for large vector datasets, as the number of sampled hierarchies is set to 1.
>
> **Comment:** How does the quality change with changes in the number of samples used?
> **Response:** We show the normalized Dasgupta cost across a different number of sampled hierarchies and edges in Fig.3. As expected, the Dasgupta cost decreases when the number of samples increases. For the edges, we can already observe an improvement towards the average linkage algorithm when sampling $\sqrt{n}$ many edges. For the number of sampled hierarchies, we observe that on most of the datasets, 20 samples are sufficient. However, even when using only one sample, EPH outperforms FPH, the second-best method:
>
>
> | Method | Brain | Citeseer | CoraML | Genes | OpenFlight | PolBlogs | WikiPhysics |
> | ------ | ----- | -------- | ------ | ----- | ---------- | -------- | ----------- |
> | FPH    | 466.94|  77.16   | 257.42 | 183.63| 355.61     |   262.48 |    538.47   |
> | EPH    | 437.12|  74.35   | 244.29 | 182.09| 332.03     |  253.39  |    494.23   |
>
> We hope that we have satisfactorily answered the questions by the reviewer. If so, we kindly ask the reviewer to consider raising their scores. We are happy to address any remaining concerns.

---

> > ### Author Response · Authors · 2023-11-21
> >
> > In light of the end of the author-reviewer discussion period on Wednesday, we would again like to highlight our response and the revised manuscript. We hope that we satisfactorily addressed your concerns. If there are any remaining questions, we would be happy to discuss them.

---

> > > ### Comment · Reviewer_gKno · 2023-11-23
> > >
> > > Thanks for the clarifications!

---

### Official Review · Reviewer_jgu2 · 2023-11-01

**Soundness:** 3 good
**Presentation:** 4 excellent
**Contribution:** 3 good
**Rating:** 8
**Confidence:** 3

**Summary:**

This paper presents methods for hierarchical clustering using gradient-based methods, in particular a optimizing the expected score (e.g., Dasgupta cost) over a distribution of tree structures.

The authors demonstrate that the proposed approach has several nice properties (e.g., global optimal corresponds to optima of discrete cost).

The proposed approach performs well empirically compared to a variety of other approaches, compared to classic approaches such as average/ward linkage agglomerative methods and other gradient-based methods..

**Strengths:**

This paper presents an interesting approach for gradient-based hierarchical clustering. Strengths include:

* **Well-Written & Thorough** - The paper is quite complete, it is a pleasure to read and provides a clear outline of the approach, provides analysis of the empirical results (e.g., Fig 3,4,5, + much of supplement), and helpful remarks about the technical details of the approach (e.g., page 5 limitations)
* **Methodological Approach** - While the parameterization of tree structures is similar to Zügner et al (2022), the details of the sampling based approach seem to be distinct and the core contribution of the paper. While these are based on existing methods, the application here is intriguing.
* **Empirical Results** - The proposed approach performs well empirically, outperforming most other methods in terms of these cost functions. There is thorough analysis which investigates the performance of the method throughout the supplemental material.

Minor:
Page 23 cuts off

**Weaknesses:**

Limitations of the paper include:
* I think that the paper could have benefited from discussion of how the proposed cost functions relate to down stream tasks of clustering (e.g., evaluation against target labels, target hierarchies, etc.) and how continuous cost functions compare to discrete ones in this setting.
* Similarly, discussion about when one prefers such methods in practice could be interesting (e.g., are there end-to-end applications?)
* More details about the line: "To obtain these for EPH and FPH, we take the most likely edge for each row in A and B, as Zügner et al. ¨ (2022) proposed"

**Questions:**

Apologies, if I have missed something, are the trees produced by EPH binary? Do you convert them into binary trees for Dasgupta cost evaluation?

---

> ### Author Response · Authors · 2023-11-17
>
> We want to thank the reviewer for their constructive and valuable feedback. We updated the manuscript based on the feedback with important text modifications highlighted in blue. In the following, we address their remarks and questions.
>
> **Comment:** How do the proposed cost functions relate to downstream tasks of clustering, and how do the continuous cost functions relate to the discrete ones?
> **Response:** To analyze the relationship between the different cost functions and downstream tasks, we performed an external evaluation, as detailed in Appendix B.2. Here, we flatten the hierarchies derived from the different methods and compare the resulting clusters with the available ground-truth labels (see Table 12).
>
> To perform an external evaluation on graph datasets, we use two synthetic HSBMs. Our goal was to answer two questions: First, how well do the inferred hierarchies align with the ground-truth hierarchies? And second, how well do the distances correlate between graphs and hierarchies? To address the first question, we calculated the Normalized Mutual information (NMI) across the different levels of the hierarchies. For the second question, we compute the Cophenetic correlation based on the shortest path and Euclidean distances of DeepWalk embeddings (see Table 4). Exp-Das and Exp-TSD almost perfectly recover the first three levels of the ground-truth hierarchy and achieve a notably high cophenetic correlation comparable to the ground truth. In both cases, Exp-TSD achieves slightly better scores, implying a better capability to recover the ground truth.
>
> **Comment:** When are such methods used in practice? E.g., are there End-to-end applications?
> **Response:** End-to-end trainable hierarchical clustering methods are particularly useful in an offline setting where results are not required in real-time. Practical use cases include jet physics and genomics [1] or clustering of products in E-commerce [2], where linkage algorithms are commonly used.
>
> **Comment:** More details about the line: "To obtain these for EPH and FPH, we take the most likely edge for each row in **A** and **B**..."
> **Response:** After the training process, EPH yields two probabilistic hierarchies, represented as matrices $\mathbf{A}\in[0,1]^{n\times n'}$ and $\mathbf{B}\in[0,1]^{n'\times n'}$. In these matrices, each row corresponds to a categorical distribution, i.e., $\mathbf{A}$ and $\mathbf{B}$ are row-stochastic matrices. As we are interested in discrete hierarchies, we convert these into discrete counterparts, $\mathbf{\hat{A}}\in\{0,1\}^{n\times n'}$ and $\mathbf{\hat{B}}\in\{0,1\}^{n'\times n'}$, respectively. While one approach is to sample an entry in each row weighted by their probability, we select the entry with the highest probability, i.e., the most likely edge. We refined this in the updated manuscript.
>
> **Comment:** Are the trees produced by EPH binary?
> **Response:** EPH allows non-binary hierarchies, providing more flexibility. While the Dasgupta cost favors binary hierarchies, TSD does not. We can observe these in our qualitative evaluation in Fig. 4. While the hierarchies inferred using Exp-TSD have multiple branches, the hierarchies inferred by Exp-Das tend to be binary.
>
> We thank the reviewer again for their comments and feedback. We hope that we have satisfactorily answered the reviewer's questions. We are happy to address any remaining concerns.
>
> [1]: Macaluso, Sebastian, Craig Greenberg, Nicholas Monath, Ji Ah Lee, Patrick Flaherty, Kyle Cranmer, Andrew McGregor, and Andrew McCallum. "Cluster trellis: Data structures & algorithms for exact inference in hierarchical clustering." In International Conference on Artificial Intelligence and Statistics, pp. 2467-2475. PMLR, 2021.
>
> [2]: Brinkmann, Alexander, and Christian Bizer. "Improving Hierarchical Product Classification using Domain-specific Language Modelling." IEEE Data Eng. Bull. 44, no. 2 (2021): 14-25.

---

> > ### Comment · Reviewer_jgu2 · 2023-11-19
> >
> > Thank you authors for your detailed and thoughtful reply.
> >
> > > Response: To analyze the relationship between the different cost functions and downstream tasks, we performed an external evaluation, as detailed in Appendix B.2. Here, we flatten the hierarchies derived from the different methods and compare the resulting clusters with the available ground-truth labels (see Table 12).
> >
> > Great! I think this analysis strengthens the paper.
> >
> > > Response: End-to-end trainable hierarchical clustering methods are particularly useful in an offline setting where results are not required in real-time. Practical use cases include jet physics and genomics [1] or clustering of products in E-commerce [2], where linkage algorithms are commonly used.
> >
> > It is not obvious to me that the jet physics applications need end-to-end methods. The generative process as I understood for the discovery of the jet reconstruction also are similar to, yet distinct from Dasgupta style costs. I believe that [1] shows only approximations to commonly used models for reconstruction that fit into the Dasgupta-like cost framework. Still I see how your method could be well suited for representing say uncertainty over of tree structures for such applications.
> >
> > I was expecting end-to-end learning of trees may matter for something where you might train an encoder of data + learn tree structure together (e.g. perhaps more like [Cattan et al, 2021](https://arxiv.org/pdf/2104.08809.pdf)?). Not asking for more experiments, just speaking out loud.
> >
> > > Response: After the training process, EPH yields two probabilistic hierarchies, ...
> >
> > Thank you, I think the updated version is improved. For further revisions, I would suggest reminding readers again in this section why taking the most likely edge is a good idea wrt to the properties of the method.
> >
> > > Response: EPH allows non-binary hierarchies, providing more flexibility. While the Dasgupta cost favors binary hierarchies, TSD does not. We can observe these in our qualitative evaluation in Fig. 4. While the hierarchies inferred using Exp-TSD have multiple branches, the hierarchies inferred by Exp-Das tend to be binary.
> >
> > Thanks yes. It is interesting that even without a strict constraint, you achieve sota Dasgupta costs. IIRC, you could improve your dasgupta costs by simply randomly partitioning any non-binary splits into binary splits? Unless I missed it (sorry), I think adding some discussion more discussion on these points would be beneficial.
> >
> > Overall, I think this is a nice and complete paper. I am considering my original score and thinking about whether or not I would like to increase it. I will post further thoughts / comments when I make such decisions.

---

> > > ### Author Response · Authors · 2023-11-20
> > >
> > > We are delighted that our responses and adaptions clarified your concerns satisfactorily.
> > >
> > > > It is not obvious to me that the jet physics applications need end-to-end methods. The generative process as I understood for the discovery of the jet reconstruction also are similar to, yet distinct from Dasgupta style costs. I believe that [1] shows only approximations to commonly used models for reconstruction that fit into the Dasgupta-like cost framework. Still I see how your method could be well suited for representing say uncertainty over of tree structures for such applications.
> > >
> > > As the reviewer mentioned, EPH might be suited to represent distributions and uncertainties of tree structures rather than a single discrete hierarchy. While we agree that the Dasgupta cost or Tree-Sampling Divergence might not be the right target metric to optimize hierarchies in jet physics, our end-to-end learnable framework allows us to optimize any differentiable metric. Non-end-to-end approaches might not optimize the final metric important to the application, e.g., average linkage in clustering, while the end-to-end optimization of EPH directly optimizes towards that metric. It would be interesting to investigate the applicability of EPH and suitable, i.e., differentiable, cost functions to jet physics, as it provides better scalability than [1], as it grows exponentially with the number of leaves. Another advantage of end-to-end learning is to stop optimization as soon as satisfactory results are achieved or a computing budget is exceeded.
> > >
> > >
> > > >I was expecting end-to-end learning of trees may matter for something where you might train an encoder of data + learn tree structure together (e.g. perhaps more like Cattan et al, 2021?). Not asking for more experiments, just speaking out loud.
> > >
> > > We agree with the reviewer that jointly encoding data and learning tree structures is an interesting use case of end-to-end learning. As shown by the end-to-end training conducted by [Chami et al., 2020](https://arxiv.org/pdf/2010.00402.pdf), learning an external model and trees jointly yields improvements over a two-step approach. In our case, we could jointly train an encoder-decoder structure in parallel with hierarchies. As we have shown that EPH successfully clusters the Cifar-100 embeddings of a ResNet-101 BiT-M-R101x1 in a two-step manner (see Table 3), it would be interesting for future work to train these in parallel.
> > >
> > > > Thank you, I think the updated version is improved. For further revisions, I would suggest reminding readers again in this section why taking the most likely edge is a good idea wrt to the properties of the method.
> > >
> > > We followed the reviewer's suggestion and added an explanation that it serves as an approximation to find the most likely hierarchy.
> > >
> > > > Thanks yes. It is interesting that even without a strict constraint, you achieve sota Dasgupta costs. IIRC, you could improve your dasgupta costs by simply randomly partitioning any non-binary splits into binary splits? Unless I missed it (sorry), I think adding some discussion more discussion on these points would be beneficial.
> > >
> > > We attribute the state-of-the-art Dasgupta costs to the end-to-end learning and alignment of the continuous and discrete cost functions. A post-processing binarization of the branches might improve the Dasgupta cost as the optimal hierarchy is binary ([Dasgupta, 2015](https://arxiv.org/abs/1510.05043) (Sec. 2.3)). If the reviewer wishes, we will include this discussion in the revised manuscript.
> > >
> > > We are happy to address any remaining questions.

---

> > > > ### Comment · Reviewer_jgu2 · 2023-11-23
> > > >
> > > > Thank you for your additional clarifications.

---

### Meta-Review · Area_Chair_JzFi · 2023-12-04

**Metareview:**

This paper proposes an approach called EPH to extend prior work from 2021 that utilizes soft hierarchy scores, instead turning to expected hierarchy scores. These exhibit differentiability and yields several computational advantages, and the method does well on the empirical study designed by the authors. While one reviewer is quite positive in their brief review, other reviewers are borderline and lean negative, with concerns about the novelty of the approach. Ultimately, there is not enough excitement that this innovation improves upon the state of the art sufficiently, as hierarchical clustering enjoys a sea of methods.

**Justification For Why Not Higher Score:**

limited novelty

**Justification For Why Not Lower Score:**

correct paper with several nice ideas

---

### Decision · Program_Chairs · 2024-01-16

Reject